⊙ | **Open Peer Review** | Parasitology | Research Article

# Identification and characterization of a carbohydrate recognition domain-like region in *Entamoeba histolytica* Gal/GalNAc lectin intermediate subunit

Hongze Zhang,[1] Qingshan Li,[1] Hang Zhou,[1] Meng Feng,[1] Yanqing Zhao,[1] Ruixue Zhou,[1] Lijun Chen,[1] Hiroshi Tachibana,[2] Xunjia Cheng[1]

**ABSTRACT** *Entamoeba histolytica* is an enteric protozoan parasite that causes human amebic colitis and extraintestinal abscesses. As a prerequisite for parasite colonization and invasion, adherence of *E. histolytica* is predominantly mediated by galactose (Gal)- and N-acetyl-d-galactosamine (GalNAc)-inhibitable lectins. The intermediate subunit (Igl) of Gal-/GalNAc-inhibitable lectin is a cysteine-rich protein containing multiple CXXC motifs and is considered a key factor affecting trophozoite's pathogenicity. However, details of the function of Igl during parasite adherence remain unclear. Here, using segmentally expressed Igl proteins and a CHO cell model transfected with Igl fragments, we identified a carbohydrate-recognition domain (CRD)-like region between amino acids 989 and 1,088. Through single- and double-point mutations in the Igl segments, two core CXXC motifs responsible for carbohydrate recognition in the CRD-like region, which are highly conserved among several lectins, were confirmed. In addition to adhesion, the roles of CRD-like region and its core CXXC motifs in various pathogenic effects were further explored. To our knowledge, this is the first report showing an adhesion-related region in *E. histolytica* Igl. The identification and characterization of this CRD-like region provides further insights into molecular mechanisms underlying *E. histolytica* pathogenicity and also aids in the determination of a potential drug target in this parasite.

**IMPORTANCE** *Entamoeba histolytica* adhesion mainly depends on galactose (Gal)-/N-acetyl-d-galactosamine (GalNAc)-inhibitable lectins, subsequently triggering a series of amebic reactions. Among the three subunits of Gal-/GalNAc-inhibitable lectin, heavy subunit and intermediate subunit (Igl) have exhibited lectin activity, but that of Igl remains poorly understood. In this study, we confirmed a carbohydrate-recognition domain (CRD)-like limiting region in *E. histolytica* Igl and further identified its two core CXXC motifs responsible for carbohydrate recognition. Moreover, the role of Igl's CRD-like region and its CXXC motifs in hemolysis and pathogenic effects was explored. This is the first study to determine an adhesion-related region in *E. histolytica* Igl protein, providing a new reference direction for subsequent research studies. Since the potential homogeneity of galectin-2 in several mammals and Igl CRD-like region, it could be meaningful to relate the corresponding pathogeneses and phenotypes of these two proteins. Except for adhesion, studies on the involvement of Igl CRD-like region in different parasite–host interactions are also promising.

**KEYWORDS** *Entamoeba histolytica*, Gal/GalNAc lectin, carbohydrate recognition domain, adhesion, CXXC motifs

*E*ntamoeba histolytica is an enteric protozoan parasite most commonly found in the tropics and subtropics, known for its remarkable ability to kill cells and cause various

Address correspondence to Xunjia Cheng, xjcheng@shmu.edu.cn, or Hiroshi Tachibana, htachiba@is.icc.u-tokai.ac.jp.

Hongze Zhang and Qingshan Li contributed equally to this article. Author order was determined by who completed the final experiments for the manuscript.

The authors declare no conflict of interest.

See the funding table on p. 19.

amebiasis (1). Though only 10% of them show severe symptoms, at least 500 million people suffer from *E. histolytica* infection worldwide, resulting in approximately 100,000 deaths from invasive amebiasis annually (2–5). Amebiasis is characterized by severe tissue destruction, which further leads to massive intestinal ulceration and sometimes even fatal extraintestinal abscesses. In the process, as an indispensable prerequisite for parasite colonization and invasion, the adherence of *E. histolytica* trophozoites to intestinal bacteria and host cells has long attracted much attention of researchers (6). On the plasma membranes of these trophozoites, a group of galactose (Gal)- and N-acetyl-D-galactosamine (GalNAc)-inhibitable lectins has been proven to be the major factor in mediating parasite adherence (7, 8).

Gal-/GalNAc-inhibitable lectin is a heterotrimeric protein complex composed of a 260-kDa heterodimer and a non-covalently associated glycosylphosphatidylinositol (GPI)-anchored intermediate subunit (Igl, 150-kDa), while the heterodimer consists of a transmembrane heavy subunit (Hgl, 170 kDa) and a GPI-anchored light subunit (Lgl, 35/31 kDa) linked by disulfide bonds (6). As a key molecule in amebic adherence, Hgl can recognize carbohydrates on the host cell surface through the cysteine-rich region between amino acids (aa) 356 and 1,143, among which the core sites are shown to be in a shorter limiting region at the C-terminal, the carbohydrate recognition domain (CRD, aa 895 to 998) (9–11). To date, only the Hgl protein of these three subunits is known to have a CRD.

With its ability of binding to a Gal-affinity column comprised of p-aminophenyl-β-D-galactopyranoside-Sepharose gel, Igl has also been identified as an important lectin protein (12). Similar to Hgl and Lgl, Igl was detected in the protein fraction binding to magnetic beads coated with GalNAc–bovine serum albumin (BSA) (13). The two isoforms of Igl, Igl1, and Igl2 are cysteine-rich proteins containing multiple CXXC motifs (14). Igl1 appears to be more closely associated with the pathogenesis of *E. histolytica* than Igl2, and their contribution to parasite adherence has been extensively studied (15, 16). Mouse monoclonal antibodies (mAb) to Igl could not only significantly block the adherence of *E. histolytica* trophozoites to human erythrocytes and Chinese hamster ovary (CHO) cells but also prevent amebic liver abscess (ALA) formation in laboratory animals (16–18). Interestingly, this adherence inhibition through the Igl protein could not be achieved by binding to an N-terminal-recognized mAb, but by binding to another C-terminal-recognized mAb (19). Igl exhibited significant hemagglutinating activity, which is mainly reflected in its C-terminal region (20). During trophozoite adherence, Igl was detected in the parasite fraction interacting with the brush border of human enterocytes (21). Under the plasma membrane of *E. histolytica* trophozoites, Igl worked with elongation factor 1 alpha to promote amebic phagocytosis (4).

However, though several studies have confirmed the contribution of Igl, details of its function during parasite adherence remain unclear. For many years, the amino acid sequences of the two Igl proteins (Igls) were generally thought to lack a CRD similar to that of other lectins (6, 14, 16). Thus, among the aforementioned uncertainties, the most critical is to determine the presence, position, and molecular properties of adhesion-related regions in the Igl protein.

In the present study, we hypothesized that Igl contains a CRD-like region because an anti-Igl (both Igl1 and Igl2) mAb, EH3077, inhibits trophozoite adherence and phagocytosis of *E. histolytica* (12). Mouse mAb EH3077 binds to Igl-C fragment, being reactive to both prokaryote- and eukaryote-expressed recombinant proteins. We aimed to confirm the presence of a CRD-like limiting region of *E. histolytica* Igl using segmentally expressed Igl proteins and a CHO cell model transfected with Igl fragments, further identifying its core CXXC motifs responsible for carbohydrate-recognizing activity. The role of Igl's CRD-like region in hemolysis and pathogenic effects was also studied.

## RESULTS

### Recombinant *E. histolytica* Igls and their CXXC motif mutant fragments

Without the signal peptide (SP) and GPI-anchored sequences, full-length Igl (aa 14–1,088) and its segments were constructed and expressed to explore adhesion-related regions (Fig. 1; Fig. S1A). According to the hydrophobicity prediction, Igl-F was divided into Igl-N, Igl-M, and Igl-C as previously described (Fig. S2A) (22). In the present study, Igl-C was further divided into Igl-C0, Igl-C1, Igl-C2, and lgl-C3 by its hydrophobic characteristics (Fig. S2B), with Igl-C0 being the overlapping fragment of Igl-M and Igl-C.

As an important special structure of *E. histolytica* Igl, key CXXC motifs in the Igl-C amino acid sequence were predicted based on three aspects: secondary structure, antigenic propensity, and protein domain (Fig. S2C through E). Predicted antigenic determinants of the Igl-C sequence are detailed in Table S1. Based on the prediction results, the most critical CXXC motif in the Igl-C0 (636–639 aa), Igl-C1 (854–857 aa), or Igl-C2 (913–916 aa) fragment was selected, and the first three critical CXXC motifs in the Igl-C3 (1,035–1,037 aa; 1,049–1,052 aa; 1,068–1,071 aa) fragment were selected. The first cysteine in each selected CXXC motif was converted into glycine (CXXC to GXXC). Single-point mutations were performed on Igl-C (Fig. S1B), while single- and double-point mutations were performed on Igl-C3 (Fig. S1C). These recombinant proteins were then used in further experiments.

### Identification of igl's CRD-like region

As a mAb that inhibits *E. histolytica* adherence and phagocytosis, characteristics of the non-reduced and reduced purified EH3077 bands are shown in Fig. S3A. After preincubation of 1 µg of EH3077 per $1 \times 10^5$ of trophozoites, erythrocytic adherence (Fig. S3B and C) and phagocytosis (Fig. S3D and E) of the parasite were significantly inhibited.

First, Igl fragments without point mutations were used to identify the presence and location of potential CRD-like regions. Based on the principle of competitive inhibition, saccharide-binding capacities of recombinant *E. histolytica* Igls were detected. Gal and GalNAc were added to confirm the types of carbohydrates identified by the Igls, with Glu used as a control. At the same molar concentration, the carbohydrate-recognizing activity of Igl protein was only manifested in segment C (Fig. 2A), while that of Igl-C was only manifested in segment C3 (Fig. 2B). Through competitive binding, Igl-C and Igl-C3 significantly reduced the fluorescence intensity caused by fluorescein isothiocyanate-labeled peanut agglutinin (FITC-PNA) combination, and this competitive process could in turn be weakened by the addition of Gal and GalNAc. Igl-C subsegments were added to the incubation environment of Jurkat cell phagocytosis by *E. histolytica* for further verification (Fig. 2C and D), and only Igl-C3 showed a significant inhibition of phagocytosis. In addition to carbohydrate-recognizing activity, we also directly examined the hemagglutinating activity of Igl-C subsegments as lectins. Three

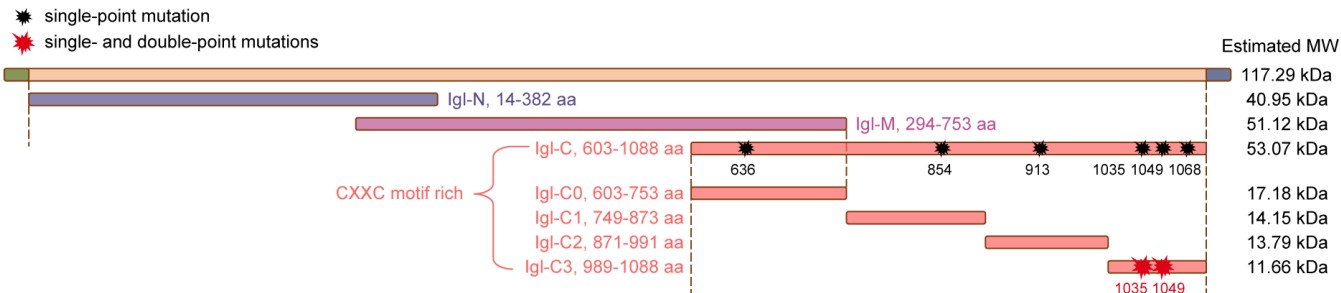

**FIG 1** Prokaryotic expression of recombinant Igl proteins used in the study. Without SP and GPI-anchored sequences, recombinant full-length Igl (GenBank accession no. AF337950.1) and its segments were constructed and expressed. Potentially key CXXC motifs in the Igl-C sequence were predicted, followed by their first cysteines being converted into glycines (CXXC to GXXC). Single-point mutations (black asterisks) were performed on Igl-C, while single- and double-point mutations (red asterisks) were performed on Igl-C3. His-tag is included in the estimated molecular weight.

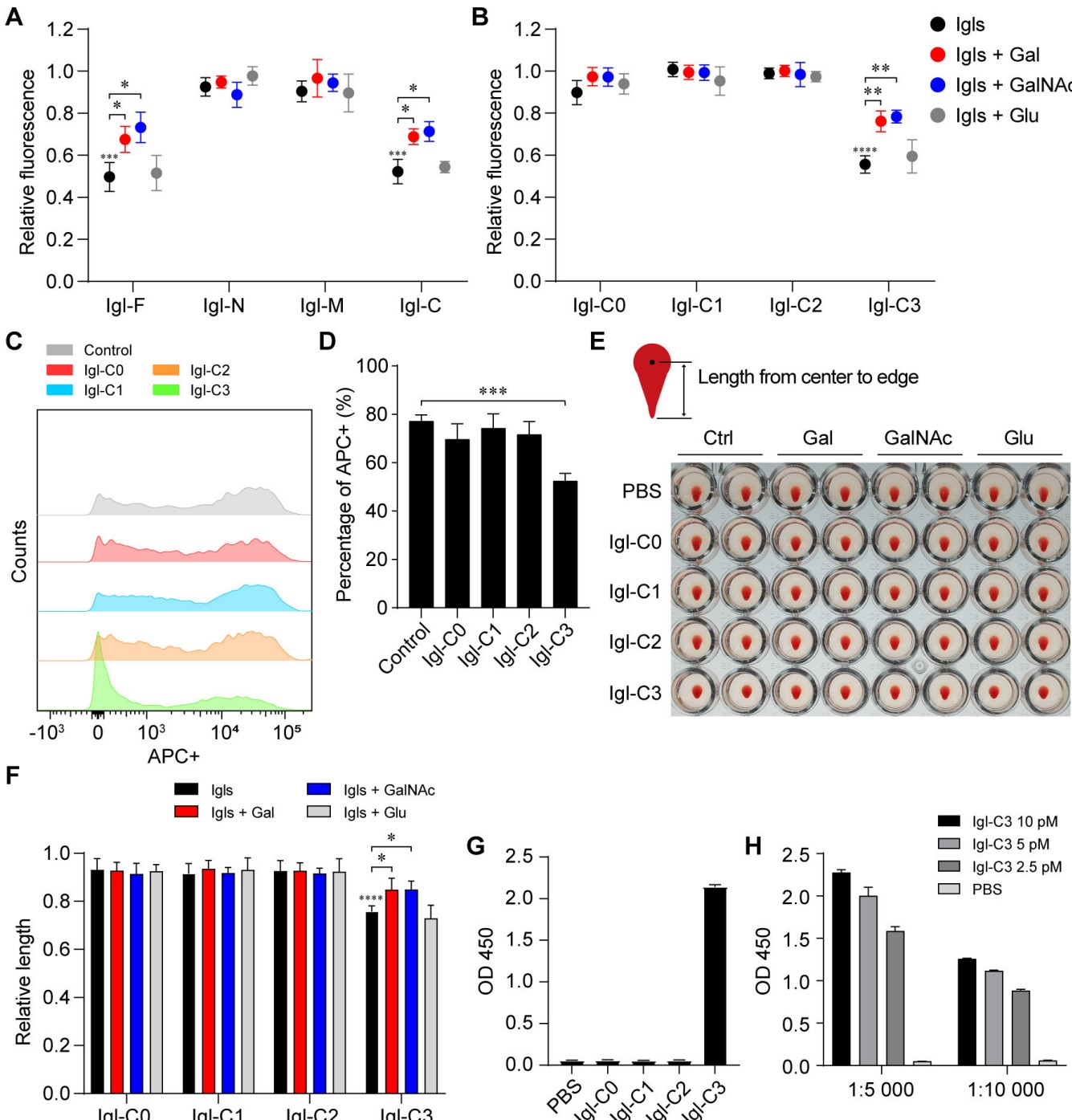

**FIG 2** Identification of the presence and location of Igl's CRD-like region. (A and B) The Igl fragments with carbohydrate-recognizing activity and the recognized carbohydrate types detected by saccharide-binding assay. Experiments were repeated three times. (C and D) Igl-C3 protein inhibits the phagocytosis of Jurkat cells (DiD labeled, APC+) by *E. histolytica* trophozoites (CFSE labeled, FITC+). Among the FITC +trophozoites, the proportion of APC +ones (due to phagocytosis) was counted. (E) Representative image of hemagglutinating effects after 1 h of incubation with different Igls. (F) Bar chart showing the lengths of deposited erythrocytes from center to edge after the microplates were tilted. Experiments were repeated three times. (G) Antigen-specific binding of Igls to EH3077 analyzed by enzyme-linked immunosorbent assay (ELISA). Coated with 10 pM protein, the dilution ratio of primary and secondary antibodies was 1:1,000 and 1:5,000, respectively. (H) Gradient results of different protein-coated amounts and secondary antibody dilutions in ELISA. Data are expressed as mean with SD. *$P$ < 0.05; **$P$ < 0.01; ***$P$ < 0.001; ****$P$ < 0.0001 by Student's *t*-test.

independent hemagglutinating assays were performed at different time points, and one of which is shown as the representative image (Fig. 2E). After adding various Igl-C subsegments, descent lengths of the deposited erythrocytes were compared with each PBS group (e.g., Igl-C3 added/Gal group and PBS/Gal group). Among the subsegments of Igl-C, only Igl-C3 exhibited obvious hemagglutinating activity, which could be inhibited by Gal/GalNAc (Fig. 2F). Only Igl-C3 in the Igl-C subsegments was reactive with EH3077 (Fig. 2G), showing a concentration-dependent manner (Fig. 2H). From the above results, the fragment with carbohydrate-recognition activity was preliminarily considered to be in the C3 segment of Igl-C.

## Core CXXC motifs for carbohydrate recognition in the CRD-like region

Six CXXC motifs in the Igl-C sequence were selected for single-point mutation. Among several mutant proteins, Igl-C-C1035G and Igl-C-C1049G exhibited not only a significant reduction in competitive binding capacity but also no response to the added Gal and GalNAc (Fig. 3A). In flow cytometry, the addition of Igl-C-C1035G or Igl-C-C1049G no longer inhibited the phagocytosis of Jurkat cells by *E. histolytica* trophozoites (Fig. 3B and C). Moreover, the hemagglutinating activities of Igl-C-C1035G and Igl-C-C1049G were also significantly reduced (Fig. 3D), and the reactivity of those of EH3077 was completely lost (Fig. 3E). To further study the role of these two CXXC motifs in carbohydrate-recognizing activity, single- and double-point mutations were performed on Igl-C3. After mutation, competitive inhibition of the proteins almost disappeared (Fig. 4A). Inhibition of the three mutant proteins on Jurkat cell phagocytosis by *E. histolytica* decreased obviously (Fig. 4B and C). The hemagglutinating activity of Igl-C3-C1035G/C1049G was decreased more dramatically than that of Igl-C3-C1035G or Igl-C3-C1049G (Fig. 4D). In the enzyme-linked immunosorbent assay (ELISA), the three mutant proteins could no longer bind to EH3077 (Fig. 4E). Concurrently, we performed flow cytometry to examine the effects of protein addition on trophozoite adherence to CHO cells. After adding Igl-C (Fig. 3F) or Igl-C3 (Fig. 4F), the positive rate of ameba adhesion was determined by calculating the ratio of Q2/(Q2 +Q3) quadrants. Probably because the FACSCalibur Flow Cytometer instrument tends to prepare each sample as single cells, the variation trend in adherence experiments was not as significant as that in phagocytosis experiments. However, though not statistically significant, the addition of Igl-C (Fig. 3G) and Igl-C3 (Fig. 4G) still led to a downward tendency in adhesion rates of CHO cells.

## Sequence alignment showing conservation of the CRD-like region and the two CXXC motifs

As another perspective to validate the importance of the two motifs (aa 1,035–1,037; aa 1,049–1,052), different amino acid sequence alignments of Igl CRD-like region were performed. Interestingly, in the alignment of Igl CRD-like region with Hgl CRD (Fig. 5A), the two CXXC motifs each contained a conserved cysteine residue (C1037 and C1049). Moreover, Igl CRD-like region was aligned with galectin-2 proteins of several mammals (Fig. 5B). Though the overall similarity among multiple sequences was not high, the above two cysteine residues were still highly conserved. Meanwhile, as a highly conserved amino acid residue in all galectins, W1046 (W65 in galectins) make an important contribution to carbohydrate recognition (23). At this site, we also found Igl CRD-like region being identical to that of several galectins.

Through the NCBI databases and sequencing of lab-cultured parasites, amino acid sequences of various *Entamoeba* Igls were obtained (see Table S2). Using *E. histolytica* Hgls as a reference, a phylogenetic tree was constructed to reveal the genetic relationships among different *Entamoeba* Igls (Fig. S4A). Subsequently, using SnapGene 6.0.2, Igls were aligned with *E. histolytica* Igl1 by the Clustal algorithm to subdivide them into N, M, C, and C0 to C3 segments. Protein BLAST (Method: compositional matrix adjust) was further used to align the same fragments between *E. histolytica* Igl1 and different *Entamoeba* Igls (e.g. *E. histolytica* Igl1-N and *E. nuttalli* Igl1-N). As shown in Table 1, the positives for Igl-C3 were always the highest in the alignments between various

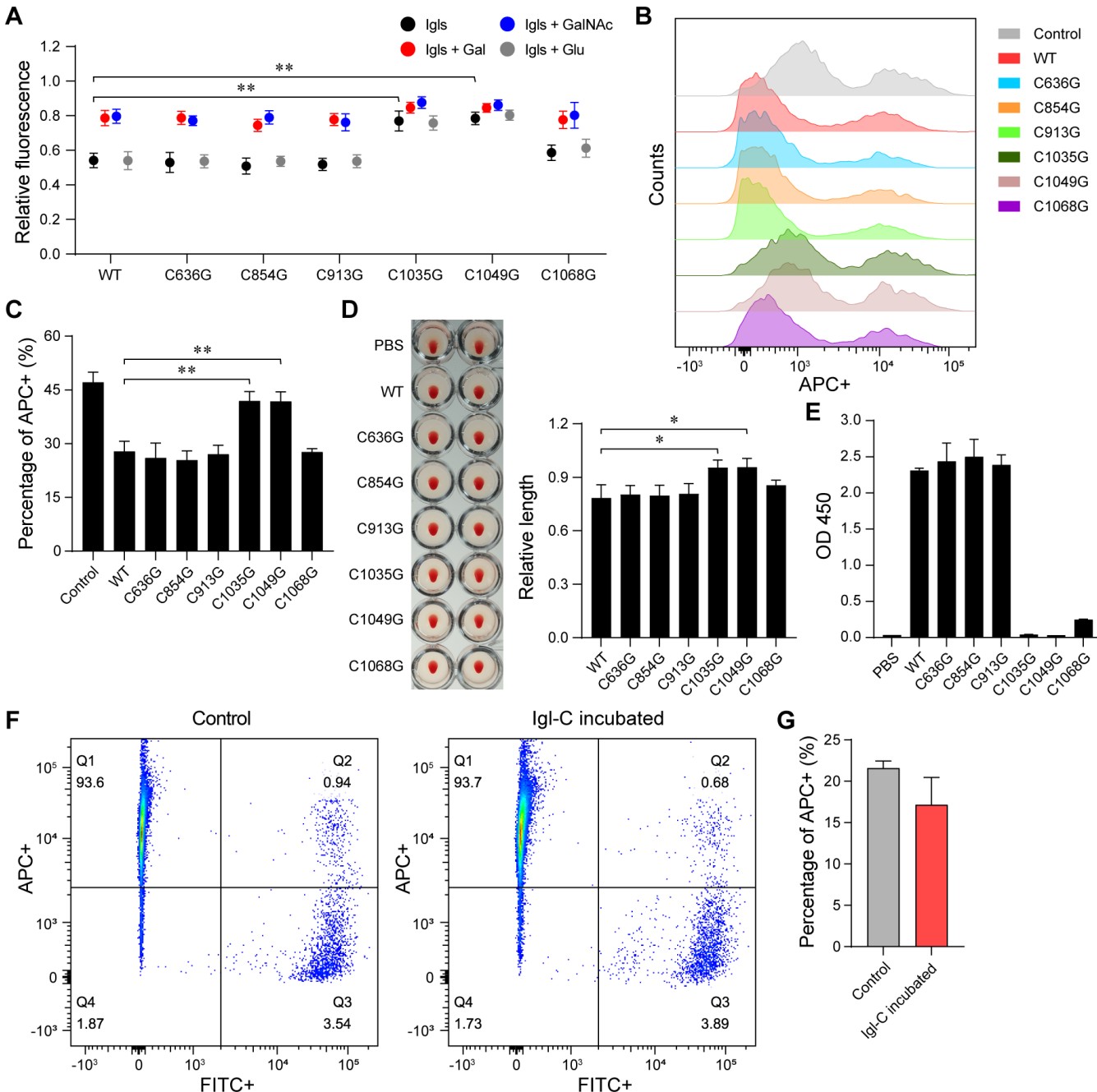

FIG 3 Mutations of several CXXC motifs in Igl-C fragment affect the function of its CRD-like region. (A) Carbohydrate-recognizing activities of Igl-C protein after single-point mutations detected by saccharide-binding assay. Experiments were repeated three times. (B and C) Inhibition of *E. histolytica* trophozoite phagocytosis of Jurkat Cells by Igl-C and its mutants. Cells were labeled with DiD (APC+), and trophozoites were labeled with CFSE (FITC+). Among the FITC +trophozoites, the proportion of APC +ones (due to phagocytosis) was counted. (D) Representative image of hemagglutinating effects after 1 h of incubation with Igl-C, and its mutants are shown. Bar chart displaying the lengths of deposited erythrocytes from center to edge after the microplates were tilted. Experiments were repeated three times. (E) Antigen-specific binding of Igl-C and its mutants to EH3077 analyzed by ELISA. (F and G) Presence of Igl-C inhibits the adherence of trophozoites to CHO cells. Likewise, cells were labeled with DiD (APC+), and trophozoites were labeled with CFSE (FITC+). Among the FITC +trophozoites, the proportion of APC +ones (due to adherence) was counted. Data are expressed as mean with SD. *$P < 0.05$; **$P < 0.01$ by Student's *t*-test.

segments, fully exhibiting its highly conserved propensity. In the sequence alignment of *Entamoeba* Igls, the conservation of CXXC motifs in each protein was compared (Fig. S4B). Only three CXXC motifs had completely positive amino acid residues (aa 395–398; aa 1,035–1,037; aa 1,049–1,052), and two of which were in segment C, being closely

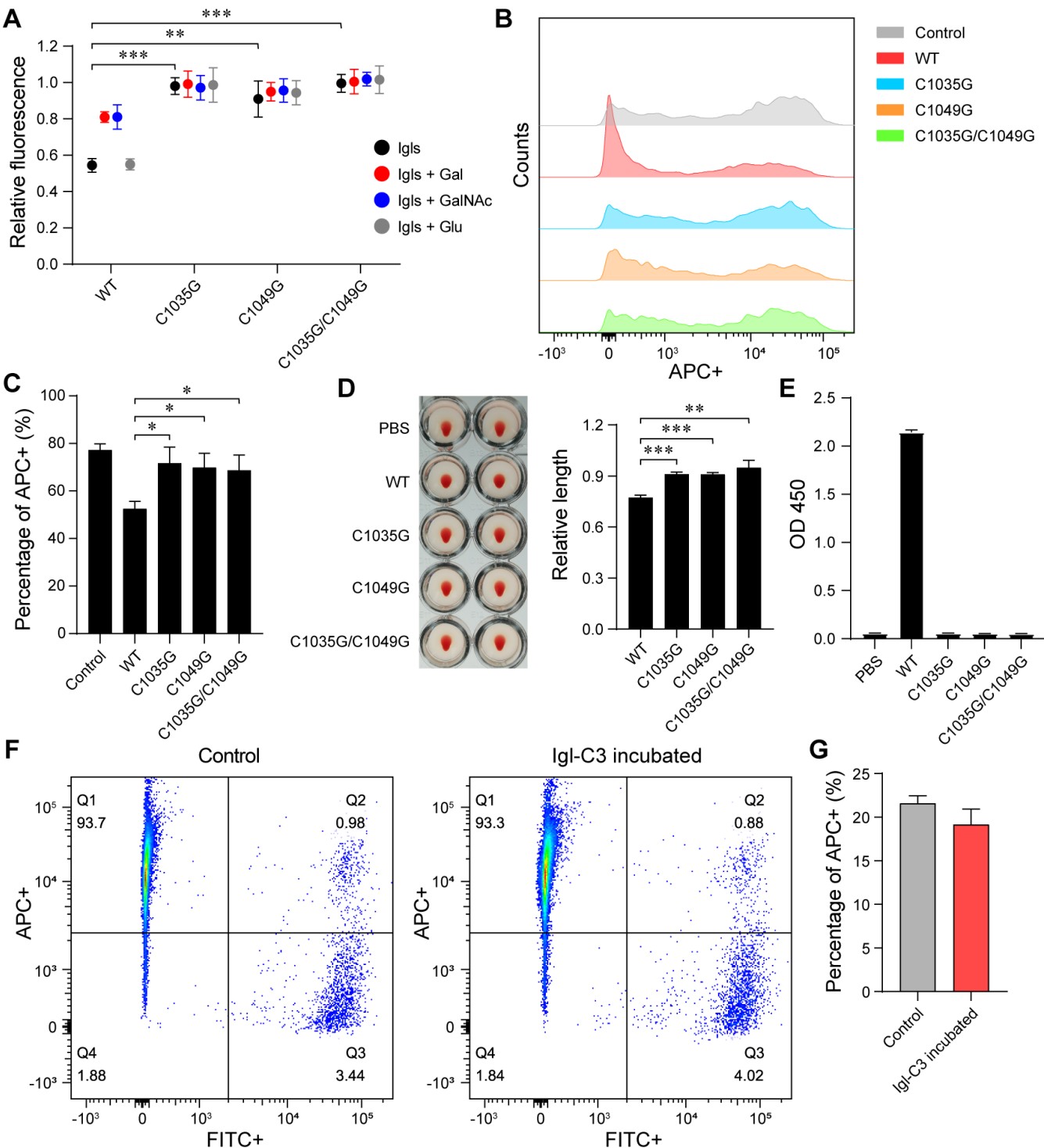

**FIG 4** Importance of the two CXXC motifs in Igl-C3 fragment for the function of its CRD-like region. (A) Carbohydrate-recognizing activities of Igl-C3 protein after single- and double-point mutations detected by saccharide-binding assay. Experiments were repeated three times. (B and C) Inhibition of *E. histolytica* trophozoite (CFSE labeled, FITC+) phagocytosis of Jurkat cells (DiD labeled, APC+) by Igl-C3 and its mutants. Among the FITC +trophozoites, the proportion of APC +ones (due to phagocytosis) was counted. (D) After 1 h of incubation with Igl-C3 and its mutants, microplates were tilted to take images of the hemagglu- tinating effect. Bar chart displaying the lengths of deposited erythrocytes from center to edge. Experiments were repeated three times. (E) Antigen-specific binding of Igl-C3 and its mutants to EH3077 analyzed by ELISA. (F and G) Presence of Igl-C3 inhibits the adherence of trophozoites (CFSE labeled, FITC+) to CHO cells (DiD labeled, APC+). Among the FITC +trophozoites, the proportion of APC +ones (due to adherence) was counted. Data are expressed as mean with SD. *$P$ < 0.05; **$P$ < 0.01; ***$P$ < 0.001 by Student's *t*-test.

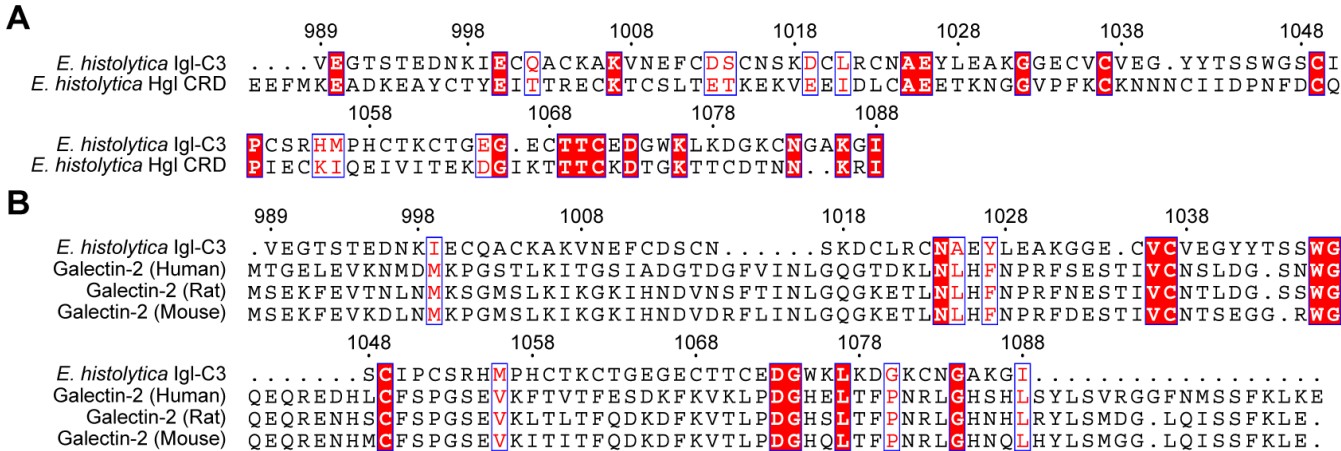

**FIG 5**  Amino acid sequence alignments of *E. histolytica* Igl CRD-like region with different CRD proteins. (A) Sequence alignment of Igl CRD-like region (UniProtKB no. Q964D2) with Hgl CRD (UniProtKB no. P23502). (B) Amino acid sequence alignment of *E. histolytica* Igl CRD-like region (UniProtKB no. Q964D2) with galectin-2 proteins in human (UniProtKB no. P05162), rat (UniProtKB no. Q9Z144), and mouse (UniProtKB no. Q9CQW5). Alignments were produced using ClustalW, and figures were generated using ESPript 3.0.

associated with carbohydrate recognition. These alignment results indicated the conserved and important role of the Igl CRD-like region and its two CXXC motifs.

## Verification by CHO cell model transfected with Igl-C3 surface proteins

After determining the importance of the Igl-C3 protein and its two CXXC motifs, a CHO cell model was constructed for verification, with a double-point mutation on the Igl-C3 sequence being performed beforehand. As shown in Fig. S5, three heterogeneously expressed fusion proteins differed in the wild type (WT), double-mutant, or empty sequence inserted between XhoI and EcoRI restriction sites. The expression of these enhanced green fluorescent protein (EGFP)–Igl–C3 fusion proteins on adherent and suspended CHO cell surfaces was examined by laser confocal microscopy (Fig. 6A). Since the antigen–antibody reaction was performed without permeabilization of detergent, fusion proteins are thought to exist on the cell surface membranes (and on membranes inside the cell). Flow cytometry (Fig. 6B) and western blotting (Fig. 6C) were also performed for cell model examination. In the hemagglutination test, CHO cells transfected with the Igl–C3–WT protein exhibited stronger hemagglutinating activity than those transfected with the control vector, while the hemagglutinating activity of Igl–C3–C1035G/C1049G-transfected cells in turn decreased compared to that of Igl–C3–WT-transfected ones (Fig. 6D). Erythrocytic adherence of the three transfected CHO cells is shown in Fig. 6E through G. The erythrocytic adhesion rate and average erythrocytic adhesion number of cells transfected with Igl–C3–WT increased significantly compared to those of control vector-transfected cells, but there was no change in the Igl–C3–C1035G/C1049G-transfected ones.

## Pathogenic effects of the CRD-like region and the two CXXC motifs

Using EH3077 as the primary antibody, immunohistochemical staining was performed to detect the distribution of *E. histolytica* Igl in the livers of ALA hamsters. During incubation, conditions were strictly controlled to be consistent between the groups, with Igl eventually stained yellow. Surprisingly, Igl was not only distributed at the site of trophozoite infection (Fig. 7A) but also widely disseminated in the surrounding healthy tissues (Fig. 7B). This suggests that Igl may have other pathogenic effects in addition to its carbohydrate-recognizing function.

To study the pathogenic effects of Igl on host cells and the role of the CRD-like region in this process, WT and double-mutant Igl-C proteins were expressed in HEK-293 cells. In the hemolytic assay of eukaryotic proteins, BSA–PBS was selected as the control

**TABLE 1** Alignment of *E. histolytica* Igl1 with various *Entamoeba* Igl segments

| Alignment | Igl-F | Igl-N | Igl-M | Igl-C | Igl-C0 | Igl-C1 | Igl-C2 | Igl-C3 |
|---|---|---|---|---|---|---|---|---|
| *E. histolytica* Igl1 | | | | | | | | |
| Identities | 100% | 100% | 100% | 100% | 100% | 100% | 100% | 100% |
| | 1,075/1,075 | 369/369 | 460/460 | 486/486 | 151/151 | 125/125 | 121/121 | 100/100 |
| Positives | 100% | 100% | 100% | 100% | 100% | 100% | 100% | 100% |
| | 1,075/1,075 | 369/369 | 460/460 | 486/486 | 151/151 | 125/125 | 121/121 | 100/100 |
| Gaps | 0% | 0% | 0% | 0% | 0% | 0% | 0% | 0% |
| | 0/1,075 | 0/369 | 0/460 | 0/486 | 0/151 | 0/125 | 0/121 | 0/100 |
| *E. histolytica* Igl2 | | | | | | | | |
| Identities | 83% | 74% | 83% | 89% | 76% | 88% | 99% | 99% |
| | 904/1,084 | 276/373 | 384/463 | 436/489 | 115/152 | 110/125 | 120/121 | 99/100 |
| Positives | 90% | 85% | 90% | 93% | 84% | 92% | 100% | 100% |
| | 978/1,084 | 318/373 | 417/463 | 456/489 | 129/152 | 115/125 | 121/121 | 100/100 |
| Gaps | 1% | 2% | 1% | 0% | 1% | 0% | 0% | 0% |
| | 14/1,084 | 8/373 | 6/463 | 3/489 | 3/152 | 0/125 | 0/121 | 0/100 |
| *E. nuttalli* Igl1 | | | | | | | | |
| Identities | 82% | 76% | 83% | 85% | 78% | 82% | 96% | 91% |
| | 893/1,084 | 284/373 | 382/463 | 417/489 | 118/152 | 102/125 | 116/121 | 91/100 |
| Positives | 89% | 85% | 88% | 90% | 84% | 88% | 96% | 97% |
| | 966/1,084 | 320/373 | 411/463 | 443/489 | 129/152 | 111/125 | 117/121 | 97/100 |
| Gaps | 1% | 1% | 1% | 0% | 1% | 0% | 0% | 0% |
| | 13/1,084 | 6/373 | 7/463 | 3/489 | 3/152 | 0/125 | 0/121 | 0/100 |
| *E. nuttalli* Igl2 | | | | | | | | |
| Identities | 83% | 77% | 82% | 89% | 81% | 88% | 95% | 97% |
| | 896/1,079 | 288/373 | 378/460 | 433/486 | 122/151 | 110/125 | 115/121 | 97/100 |
| Positives | 91% | 87% | 90% | 95% | 91% | 95% | 97% | 99% |
| | 982/1,079 | 326/373 | 417/460 | 463/486 | 138/151 | 119/125 | 118/121 | 99/100 |
| Gaps | 0% | 1% | 0% | 0% | 0% | 0% | 0% | 0% |
| | 6/1,079 | 4/373 | 2/460 | 1/486 | 1/151 | 0/125 | 0/121 | 0/100 |
| *E. dispar* Igl1 | | | | | | | | |
| Identities | 76% | 74% | 78% | 78% | 76% | 72% | 85% | 77% |
| | 830/1,089 | 277/373 | 358/461 | 384/495 | 115/151 | 94/131 | 102/120 | 77/100 |
| Positives | 86% | 85% | 87% | 87% | 87% | 81% | 92% | 92% |
| | 944/1,089 | 318/373 | 404/461 | 435/495 | 132/151 | 107/131 | 111/120 | 92/100 |
| Gaps | 1% | 2% | 1% | 2% | 2% | 4% | 0% | 0% |
| | 19/1,089 | 8/373 | 5/461 | 10/495 | 4/151 | 6/131 | 0/120 | 0/100 |
| *E. dispar* Igl2 | | | | | | | | |
| Identities | 75% | 68% | 78% | 77% | 72% | 73% | 84% | 78% |
| | 809/1,085 | 252/370 | 357/460 | 380/495 | 109/151 | 96/131 | 101/120 | 78/100 |
| Positives | 85% | 81% | 88% | 87% | 86% | 83% | 90% | 93% |
| | 933/1,085 | 301/370 | 408/460 | 433/495 | 130/151 | 109/131 | 108/120 | 93/100 |
| Gaps | 1% | 1% | 0% | 1% | 1% | 4% | 0% | 0% |
| | 15/1,085 | 5/370 | 4/460 | 9/495 | 3/151 | 6/131 | 0/120 | 0/100 |
| *E. moshkovskii* Igl-like | | | | | | | | |
| Identities | 41% | 37% | 40% | 44% | 37% | 41% | 49% | 60% |
| | 452/1,116 | 141/383 | 189/470 | 221/500 | 58/155 | 51/125 | 58/119 | 55/92 |
| Positives | 55% | 53% | 54% | 56% | 47% | 54% | 62% | 76% |
| | 614/1,116 | 204/383 | 254/470 | 284/500 | 74/155 | 68/125 | 74/119 | 70/92 |
| Gaps | 7% | 12% | 7% | 4% | 7% | 6% | 0% | 0% |
| | 88/1,116 | 49/383 | 35/470 | 22/500 | 12/155 | 8/125 | 1/119 | 0/92 |
| *E. invadens* Igl-like | | | | | | | | |
| Identities | 33% | 30% | 30% | 37% | 33% | 34% | 33% | 59% |
| | 362/1,113 | 118/388 | 140/470 | 179/488 | 47/144 | 41/122 | 39/117 | 51/87 |

*(Continued on next page)*

**TABLE 1** Alignment of *E. histolytica* Igl1 with various *Entamoeba* Igl segments (*Continued*)

| Alignment | Igl-F | Igl-N | Igl-M | Igl-C | Igl-C0 | Igl-C1 | Igl-C2 | Igl-C3 |
|---|---|---|---|---|---|---|---|---|
| Positives | 47% | 44% | 47% | 50% | 47% | 46% | 47% | 72% |
| | 525/1,113 | 173/388 | 222/470 | 247/488 | 69/144 | 57/122 | 55/117 | 63/87 |
| Gaps | 10% | 10% | 8% | 12% | 9% | 13% | 17% | 2% |
| | 122/,1113 | 41/388 | 38/470 | 61/488 | 13/144 | 17/122 | 21/117 | 2/87 |

because PBS alone can also cause obvious hemolysis (Fig. 7C). Igl–C–WT exhibited significant hemolytic activity, which was inhibited by Gal and GalNAc. In contrast, the hemolytic activity of Igl–C–C1035G/C1049G was significantly reduced compared to that of Igl–C–WT, but did not disappear completely. Likewise, transfected CHO cells were used for hemolytic assays (Fig. 7D). CHO cells transfected with the Igl–C3–WT protein exhibited significant hemolytic activity, while that of Igl–C3–C1035G/C1049G-transfected cells decreased dramatically. Caco-2 cell viability was detected to reflect the cytotoxicity of the CRD-like region (Fig. 7E). At all time-points, cell viability of the Igl–C–WT incubated group decreased significantly. As expected, cell viability of the Igl–C–C1035G/C1049G incubated group rose again compared to that of the Igl–C–WT incubated one. We previously found that *E. histolytica* infection could lead to inflammatory reactions such as increased secretion of proinflammatory cytokines and macrophage polarization (24). In the present study, to quantify the corresponding changes caused by Igl CRD-like region, qRT-PCR was used to amplify interleukin (IL)-1β, IL-6, tumor necrosis factor alpha (TNF-α), and inducible nitric oxide synthase (iNOS) genes in RAW264.7 cells (Fig. 7F). GAPDH was amplified as a reference. At 12 and 24 h, expression levels of the four genes in cells preincubated with Igl–C–WT were dramatically increased, while those in Igl–C–C1035G/C1049G preincubated cells exhibited varying degrees of reduction.

## DISCUSSION

In *E. histolytica* infection, the adhesion of trophozoites to intestinal bacteria and host cells mainly depends on Gal/GalNAc lectins, subsequently triggering a series of amebic reactions such as the release of cysteine proteases, amebapores, and peroxiredoxin (6, 22, 25–28). Among the three subunits of Gal-/GalNAc-inhibitable lectins, Hgl and Igl have exhibited lectin activities, but that of Igl remains poorly understood. Commercial glycan arrays were used to check the lectin activity of Igl1, failing to detect regions with significant affinity (20). However, the affinity of Hgl for single GalNAc residues was found to be lower than that of multiple GalNAc-attached neoglycoproteins, and Igl may therefore also exhibit a weak affinity for glycans on the array (29, 30). In the present study, referring to the research on *Fusobacterium nucleatum* adhesion to cancer cells, we performed saccharide-binding assays to determine the carbohydrate-recognizing activity of various Igl segments (31). Since glycoproteins on the surface of intestinal tumor cells possess more Gal/GalNAc modifications, Caco-2 cells were used in this experiment (31, 32). Through competitive binding between Igl segments and FITC-PNA and further addition of Gal/GalNAc, we identified the presence of a CRD-like region in Igl–C3. Similar to Hgl CRD (aa 895–998), a limiting region of approximately 100 amino acids (Igl CRD-like region, aa 989–1,088) is sufficient for adhesion (33). EH3077 binding tests, trophozoite adherence, and phagocytosis inhibition tests were performed for verification from different perspectives, while hemagglutinating tests were also performed to directly examine lectin activity.

One-eighth of the amino acid residues in Igl are cysteines, most of which are in the form of CXXC motifs located in the M and C segments (14). CXXC motif-containing proteins are involved in various biological processes, including redox activity, transcriptional regulation, cell proliferation, and development (34–36). Identified with different functions, the structure of CXXC motifs plays an important role in multiple protozoan parasites. In *Plasmodium* and *Trypanosoma*, the conserved CXXC motifs contribute to the maintenance of oxidoreductase activity, potentially leading to persistent infection (37–39). In *Giardia lamblia*, similar CXXC motifs are associated with protein–

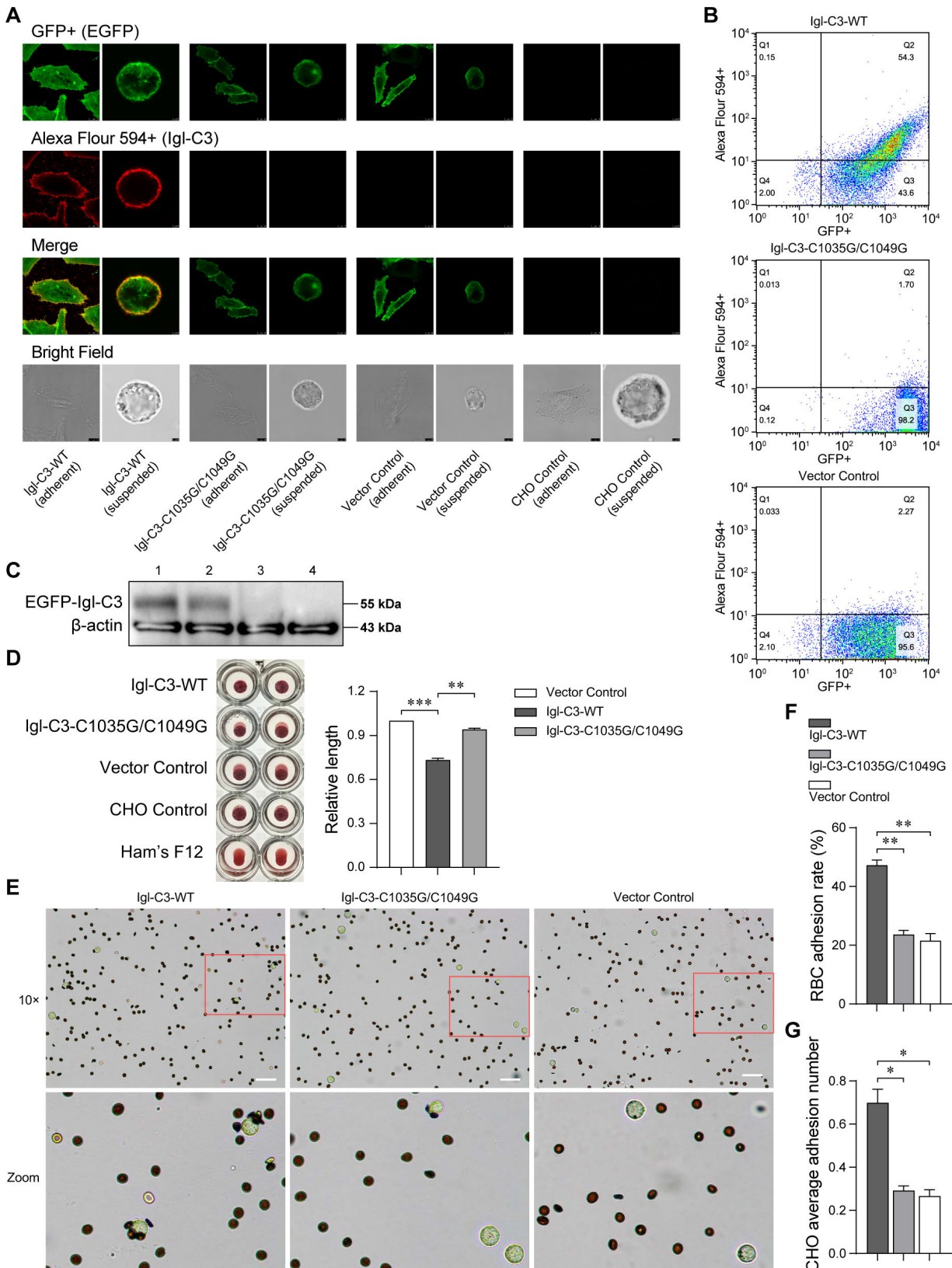

**FIG 6** The role of Igl's CRD-like region and its key CXXC motifs was verified by transfected CHO cells. (A) Laser confocal microscopy was used to determine the expression of wild-type, double-mutant, and vector control EGFP–Igl–C3 proteins on the adherent or suspended CHO cell surface, with EGFP detected by GFP fluorescence and Igls detected by Alexa Flour 594 fluorescence (EH3077 as the primary antibody). (B) CHO cell transfection examined by flow cytometry. (Continued on next page)

Fig 6 (Continued)

EGFP was detected by GFP fluorescence, and Igls were detected by Alexa Flour 594 fluorescence (EH3077 as the primary antibody). (C) CHO cell transfection examined by western blotting. The expression and band location of EGFP–Igl–C3 fusion proteins were detected by GFP mAb. In turn, lanes 1 to 4 are Igl–C3–WT, Igl–C3–C1035G/C1049G, Vector Control, and CHO Control groups. (D) Image of hemagglutinating effects after 8 h of incubation with transfected CHO cells is shown. Bar chart displaying the lengths of deposited erythrocytes from center to edge. (E) Erythrocytic adherence of CHO cells transfected with wild-type and double-mutant Igl–C3 surface proteins. Images were taken with a ×10 objective lens and zoomed in locally. Scale bar: 100 µm. (F and G) The RBC adhesion rate and average RBC adhesion number per cell of several transfected CHO cells were counted, respectively. $*P < 0.05$; $**P < 0.01$; $***P < 0.001$ by Student's $t$-test.

protein interactions and variable antigenicity (14, 22). Since Hgl CRD is also located in the cysteine-rich region, we investigated the influence of different cysteine residues and CXXC motifs on Igl's carbohydrate-recognizing activity (11). Through a series of experiments using mutant Igl proteins, two core motifs (aa 1,035–1,037; aa 1,049–1,052) were identified in the CRD-like region. Owing to the ability of accumulating polyploid cells in their proliferative phase, gene silencing of WT Igl and alternative expression of mutant Igls in *E. histolytica* trophozoites are difficult to achieve (40). In the present study, a CHO cell model transfected with different Igl–C3 surface proteins was therefore constructed to validate the above results.

In addition to biological experiments, we further performed sequence alignments of Igl CRD-like region and multiple lectins. As a proto-type galectin, galectin-2 is abundant in various tissues, such as the gastrointestinal tract, placenta, and cardiovascular system, being involved in different physiological and pathological processes (41–44). Altered expression of galectin-2 has been implicated in the pathogenesis of inflammatory bowel disease (IBD), pregnancy-related disorders, and several cancers (45, 46). Interestingly, in the sequence alignments with galectin-2 proteins of several mammals and the Hgl CRD, the two core CXXC motifs in Igl CRD-like region each contain a highly conserved cysteine residue. In our experiments, mutations in these two motifs led to a dramatic reduction in carbohydrate-recognition activity. The two cysteine residues, C57 (C1037 in Igl CRD-like region) and C75 (C1049 in Igl CRD-like region), are also highly conserved in galectin-2 proteins and have important functions (42). Galectin-2 can lose its carbohydrate binding and hemagglutinating ability via oxidative inactivation, while C57 has been identified to be responsible for controlling hydrogen peroxide-induced inactivation (42, 47). Likewise, the C75 mutation directly leads to a significant reduction in lectin activity (48). These highly conserved features suggest potential homogeneity of Igl CRD-like region and galectin-2 proteins.

From another perspective, we aligned the same fragments between *E. histolytica* Igl1 and different *Entamoeba* Igls to verify the conservation of the CRD-like region and its core CXXC motifs. Among all Igl subsegments, Igl–C3 is the most conserved among different *Entamoeba* species. Only three CXXC motifs were completely positive among *Entamoeba* Igls, two of which were core motifs in the CRD-like region. Coincidentally, the Hgl CRD sequence is not only completely conserved among members of the *E. histolytica* lectin family, but 89% conserved in homologous proteins expressed by *Entamoeba dispar* (10). Thus, these highly conserved results suggest a potentially vital role for the CRD-like region and the two CXXC motifs.

Over the years, other than trophozoite attachment, the pathogenic effects of *E. histolytica* Igl on host cells have been studied extensively. Igl has not only strong immunogenicity but also significant hemolytic and cytotoxic activities (20, 22). Igl is also highly involved in parasite–host interactions, which can promote the virulence of *E. histolytica* by regulating the central carbon metabolism of host cells (49). In the immunohistochemical staining using livers from ALA hamsters, we found that Igl was widely disseminated in healthy tissues surrounding the site of *E. histolytica* infection. Probably due to the detachment from the membrane surface or trophozoite rupture, Gal/GalNAc lectins can be transferred from the parasite to the surface of host cell membranes, which confirms the functional basis of Igl's pathogenic effects in addition to adhesion (20, 50, 51). In the present study, to avoid interference from endotoxins and

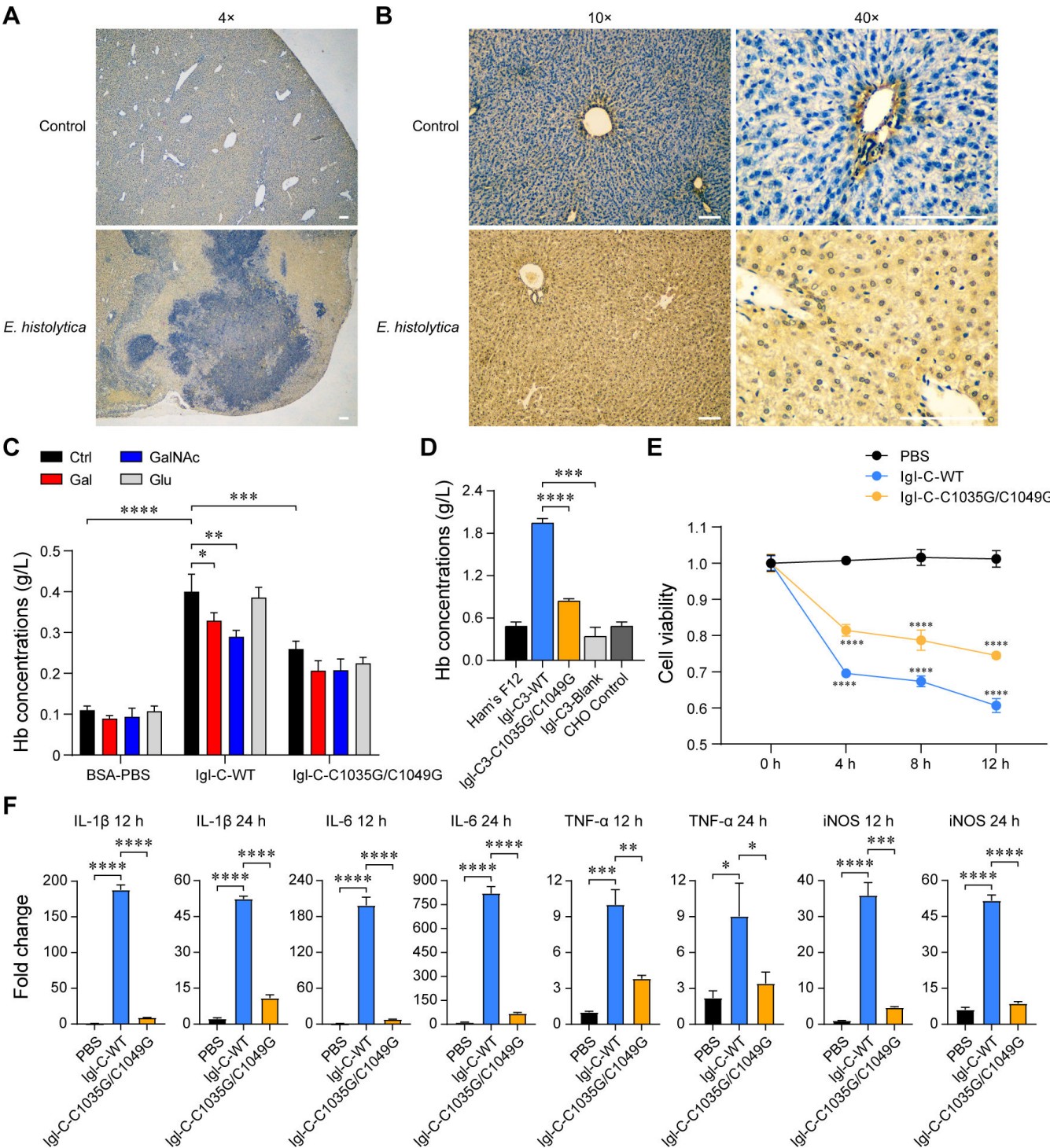

**FIG 7** Dissemination of Igl protein in *E. histolytica* infection and the pathogenic effect of its CRD-like region on host cells. (A) Representative immunohistochemical images showing the distribution of Igl in liver tissue from ALA hamster model. The trophozoite infection site is shown. Scale bar: 100 µm. (B) Healthy liver tissue surrounding the infection site is shown. Scale bar: 100 µm. (C) Hb concentrations released from the supernatant after incubation with Igl-C proteins and monosaccharides in the hemolytic assay. (D) Hb concentrations released from the supernatant after incubation with transfected CHO cells. (E) Line chart showing Caco-2 cell viabilities determined by the quantitative detection of ATP. The significance on each blue node is generated by comparing the corresponding node on the black line, while the significance on each yellow node is generated by comparing that on the blue line. (F) qRT-PCR assay of IL-1β, IL-6, TNF-α, and iNOS expressions in RAW264.7 cells preincubated with Igl-C proteins. *$P < 0.05$; **$P < 0.01$; ***$P < 0.001$; ****$P < 0.0001$ by Student's *t*-test.

the lack of post-translational modifications, eukaryotic Igl–C proteins were expressed for experiments. Likewise, since we previously used Igl–C for a series of studies about serodiagnosis, immunogenicity, and protein–protein interactions, pathogenic effects of the CRD-like region and the two CXXC motifs on host cells were confirmed using Igl–C instead of Igl–C3 proteins (4, 16, 22). We found that the hemolytic activity of CHO cells transfected with Igl–C3 decreased dramatically after mutation, while that of eukaryotic Igl–Cs remained at a high level despite the significant decrease after mutation. As the first target of *E. histolytica* destruction in the host, the human intestinal epithelial cells, Caco-2, were selected to detect the cytotoxicity of Igl–C proteins. Though cytotoxicity was reduced after Igl–C mutation, it was also maintained at a certain level. In the Igl–C segment, three regions (aa 787–846; aa 968–1,028; aa 1,029–1,088) have been demonstrated with hemolytic and cytotoxic activities, which can be partially inhibited by Gal and GalNAc (52). Owing to the presence of multiple separated active regions, it is reasonable for Igl–C–C1035G/C1049G to retain part of its hemolysis and cytotoxicity. In our previous study on host inflammatory immune responses following the interaction with *E. histolytica* parasites, the macrophage polarization is found to skew toward the M1 phenotype, with the expression levels of iNOS (a marker of M1 macrophage polarization), IL-1β, IL-6, and TNF-α (several representative proinflammatory cytokines) being significantly increased (24). Here, to determine the role of Igl CRD-like region and its core CXXC motifs in this process, the expression of corresponding genes was detected by qRT-PCR. Apparently, the CRD-like region contributes to (but not all) the M1 macrophage polarization caused by trophozoites to a certain extent.

This study did have some limitations. On the one hand, most of the Igl segments used in our experiments were prokaryote expressed. Though expression products of the soluble portion were selected for purification to ensure the correct structure of these proteins, such as disulfide bonds, the lack of many post-translational modifications in the prokaryotic system remains. Thus, it is meaningful to conduct further researches using native *E. histolytica* Igl and eukaryote-expressed Igl fragments. On the other hand, most of the methods used in this study to determine the Igl CRD-like region were traditional biological experiments. We attempted to detect the structure of native *E. histolytica* Igl to identify its potential carbohydrate-binding sites, but without success. This may be due to the properties of such membrane proteins and the complex spatial structure caused by disulfide bonds of excessive cysteines, which also results in Igl protein's low model confidence in the AlphaFold structure prediction (53). Meanwhile, the lack of similarity between Igl protein and lectins of other species increases the difficulty of structural prediction and analysis. Much efforts are still needed to overcome these difficulties.

To our knowledge, this is the first study to determine an adhesion-related region in *E. histolytica* Igl protein. Two highly conserved CXXC motifs were found to be responsible for the carbohydrate-recognizing activity of Igl's CRD-like region. Moreover, in addition to adhesion, we identified the role of CRD-like region and its core CXXC motifs in various pathogenic effects. These features provide further insights into molecular mechanisms underlying *E. histolytica* pathogenicity, and also aid in the identification of a potential drug target for the parasite. However, much work remains to be done to consummate the understanding of the Igl CRD-like region. Since the carbohydrate-binding activity of Hgl CRD can be calcium-dependent and has a calcium-bound DPN motif in its sequence, it is necessary to determine the effect of calcium on the corresponding activity of the Igl CRD-like region (10, 33). Due to the potential homogeneity of the Igl CRD-like region and galectin-2, it would be meaningful to relate the corresponding pathogeneses and phenotypes of these two proteins. For example, exogenous galectin-2 has been found to be associated with gastrointestinal wound healing and integrity of the epithelial architecture in IBD, while amebic colitis and IBD not only have similar clinical manifestations but also occur as relatively frequent co-infections (44, 54–58). Except for adhesion, the involvement of Igl CRD-like region in parasite–host interactions also deserves further elaboration. Our previous study demonstrated that purified Igl protein can induce metabolic reprogramming in host cells, leading to a Warburg-like effect similar to that

observed in cancers (49). Likewise, we previously found the M1 macrophage polarization caused by *E. histolytica* incubation, while the present study exhibited that the Igl CRD-like region seemed to play an important role within this process (24). It would be quite interesting to explore the pathogenicity of the Igl CRD-like region in different host cells and its molecular mechanisms.

## MATERIALS AND METHODS

### Cloning and plasmid generation of Igls and the mutants

Prokaryotic plasmids were constructed as previously described (49). In brief, as full-length Igl (GenBank accession no. AF337950.1) without the SP and GPI-anchored sequences, Igl–F and its segments were amplified and subcloned into pET-19b vectors (Novagen, Darmstadt, Germany) within the XhoI restriction site. After obtaining the EGFP sequence from a pWPXL vector (Addgene, #12257), FLAG–HA–pcDNA3.1-vectors (Addgene, #52535) were used for EGFP–Igl–C3 expression on the CHO cell surface. First, target sequences were synthesized by overlapping PCR. SP (hamster) was inserted between NheI and XhoI restriction sites, linker–EGFP–GPI (hamster) was inserted between BamHI and HindIII restriction sites, while Igl–C3 was inserted between XhoI and EcoRI restriction sites. For eukaryotic expression of the Igl–C protein in HEK-293 cells, the FLAG–linker–Igl–C sequence was obtained by overlapping PCR, then subcloned into a pCMV6-Entry vector (OriGene, #PS100001) within SgfI and MluI restriction sites. Overlapping PCR was also used for the generation of all relevant mutants. The primer sets are shown in Table S3.

### Expression and purification of recombinant proteins

The prokaryotic expression and purification of Igl proteins were as described previously (59). After being transformed with the subcloned plasmids, *E. coli* BL21 Star(DE3)pLysS competent cells (Weidibio, Shanghai, China) were cultured in Luria–Bertani medium containing ampicillin (100 µg/mL), then induced with isopropyl-β-d-thiogalactopyranoside (Amresco, Solon, OH, USA) for recombinant protein expression at a final concentration of 1 mM. The bacteria were incubated at 37°C for 3 h, then harvested and sonicated. The supernatants were filtrated via a 0.45-µm membrane, further purified using a Ni-NTA (nitrilotriacetic acid) His-Bind resin kit (Novagen, Darmstadt, Germany). Purity of the protein samples was confirmed by sodium dodecyl sulfate-polyacrylamide gel electrophoresis (SDS-PAGE) and Coomassie Brilliant Blue staining (see Fig. S1). A Toxin Sensor Gel Clot Endotoxin Assay Kit (Genscript, Nanjing, China) was applied to detect the endotoxin levels in the recombinant proteins, which conformed to the national standard of the People's Republic of China for medical products (GB/t14233.2–2005).

After amplification within *E.coli* JM109 competent cells (TaKaRa, Shiga, Japan), vectors for eukaryotic transfection were extracted with an EndoFree Plasmid Maxi Kit (Qiagen, Dusseldorf, Germany). According to the manufacturer's instructions, CHO cells were transfected using a Lipofectamine 3000 Reagent (Invitrogen, L3000015; Carlsbad, CA, USA), while HEK-293 cells were transfected by the PEI method. The transfected cells were incubated at 37°C for 48 h, then screened with geneticin-selective antibiotic (G418 sulfate; Invitrogen, 10131035). The Igl–C protein expressed in HEK-293 cells was purified through a FLAG tag, then quantified and confirmed using SDS-PAGE.

### Amebic strains and cell cultures

Trophozoites of *E. histolytica* isolates, HM-1:IMSS and SAW755CR, were grown axenically at 36.5°C, in BI-S-33 medium containing 10% and 15% heat-inactivated adult bovine serum (Sigma-Aldrich, St. Louis, MO, USA), respectively. CHO-K1, Jurkat Clone E6-1, HEK-293, and RAW264.7 cells were separately grown in Ham's F-12 medium, RPMI-1640 medium, Eagle's minimum essential medium (MEM), and Dulbecco's modified Eagle's medium (Corning, Manassas, VA, USA) supplemented with 10% fetal bovine serum

(HyClone Laboratories, Logan, UT, USA). Human intestinal epithelial cells, Caco-2, were grown in MEM medium supplemented with 20% fetal bovine serum, MEM non-essential amino acids (Gibco, Grand Island, NY, USA), 2 mM L-glutamine (Gibco, Grand Island, NY, USA), and 1 mM sodium pyruvate (Gibco, Grand Island, NY, USA). Cells were grown at 37°C in a 5% $CO_2$ incubator.

## Saccharide-binding assay

Assays to detect the binding ability of recombinant *E. histolytica* Igls to multiple saccharides were conducted in 96-well tissue-culture treated microplates with black well walls and clear bottoms (PerkinElmer, 6005182; Waltham, MA, USA). First, 100 µL of Caco-2 cells ($3 \times 10^4$) was spread into each well and incubated for 24 h. Cells were treated with recombinant proteins (300 nM) and monosaccharides (60 mM) in serum-free MEM medium, incubated on ice for 90 min, then washed three times with ice-cold PBS. Subsequently, FITC-PNA (150 nM; Sigma-Aldrich, L7381; St. Louis, MO, USA), a Gal-/GalNAc-specific lectin, was added into wells, followed by incubation on ice in the dark for 90 min. After washing with PBS, the fluorescence intensities (490-nm excitation; 525-nm emission) were measured on a BioTek Synergy H1 Microplate Reader (Agilent BioTek, Vermont, USA).

## Hemagglutinating and hemolytic assays

Assays were performed with erythrocytes from human blood group O using 96-well round-bottom microplates (Corning 3799; Kennebunk, USA). In hemagglutination tests with various recombinant *E. histolytica* Igls, 50 µL of proteins (4 µM) and monosaccharides (120 mM) were first loaded into each well. Erythrocytes, 2% (vol/vol), were prepared in PBS, and 50 µL was added into the preloaded wells. After 1 h of incubation at room temperature, the microplates were tilted, then the lengths of deposited erythrocytes were measured from center to edge to assess hemagglutinating activities (20). The hemagglutination test of transfected CHO cells was conducted in serum-free Ham's F-12 medium. After adding 100 µL of cells ($2 \times 10^6$) and 100 µL of 6% (vol/vol) erythrocytes into each well, the microplates were incubated at 37°C for 8 h.

For the hemolytic assay, 50 µL of erythrocytes ($5 \times 10^7$) was added into wells containing 50 µL of proteins (2 µM) and monosaccharides (120 mM), followed by incubation at 37°C for 1.5 h. Alternatively, 100 µL of transfected CHO cells ($2 \times 10^6$) were prepared in serum-free Ham's F-12 medium, added into wells preloaded with 100 µL of 6% (vol/vol) erythrocytes, then incubated at 37°C for 12 h. The quantification of hemolytic activities was performed by measuring the concentration of hemoglobin (Hb) in the supernatants above with a Free Hemoglobin Assay Kit (Jiancheng, Nanjing, China).

## Erythrocytic adherence and phagocytosis

Mouse mAb EH3077 was previously prepared (12). For the erythrocytic adherence and phagocytosis assays of *E. histolytica* trophozoites, amebae ($1 \times 10^5$) were preincubated with different concentrations of purified EH3077 antibodies or equivalent BSA (GPC, AA691; Beijing, China) at 37°C for 30 min. In 1% adult bovine serum BI-S-33 medium, trophozoites were incubated with erythrocytes (1:100 ratio) on ice for 5 min to detect adhesion levels. For the erythrophagocytosis experiments, trophozoites and erythrocytes (1:100 ratio) were incubated at 37°C for 5 min, then distilled water was quickly added to dissolve the free erythrocytes. After 2.5% glutaraldehyde fixation, diaminobenzidine (DAB) of 1 mg/mL was used for erythrocyte staining. Adherence and phagocytosis rates were determined by examination of at least 300 trophozoites, and amebae with at least three erythrocytes attached were considered positive for adhesion.

To detect the erythrocytic adherence of CHO cells transfected with WT and double-mutant Igl–C3 surface proteins, cells and erythrocytes (1:50 ratio) were prepared in serum-free Ham's F-12 medium, incubated on ice for 1 h, then fixed with 2.5% glutaralde-

hyde, and stained with DAB. Of the 300 CHO cells examined, those with at least three adherent erythrocytes were considered positive.

## Immunoreactivity measurement by ELISA

The antigen-specific binding of EH3077 to various recombinant *E. histolytica* Igls was analyzed by ELISA. After being coated with Igls (10 pM), ELISA microplates (Corning 92592; Kennebunk, USA) were blocked with PBS containing 1% skim milk. EH3077 (1:1,000) and horseradish peroxidase (HRP)-conjugated goat anti-mouse IgG H&L (1:5,000; Abcam, Ab6789; Cambridge, United Kingdom) were used as primary and secondary antibodies, respectively. With a TMB-Elisa liquid substrate solution (GPC, Beijing, China), optical density (OD) was measured at 450 nm after the incubation at room temperature for 10 min.

## Flow cytometry

For the detection of *E. histolytica* phagocytosis and adherence, Jurkat and CHO cells were first incubated with 10 µM 1,1′-dioctadecyl-3,3,3′,3′-tetramethylindodicarbocyanine, 4-chlorobenzenesulfonate salt (DiD) at 55°C for 20 min to label the cells and induce cell death, while trophozoites were labeled with 20 µM carboxyfluorescein succinimidyl ester (CFSE) at room temperature for 10 min. For the phagocytic assay, trophozoites ($1 \times 10^5$) were co-incubated with DiD-labeled heat-killed Jurkat cells ($1 \times 10^6$, 1:10 ratio) in the presence of 10 µM of various recombinant *E. histolytica* Igls at 37°C for 20 min, then fixed with 4% paraformaldehyde. The adherence of *E. histolytica* was measured as previously described (12). Trophozoites ($5 \times 10^4$) and DiD-labeled heat-killed CHO cells ($1 \times 10^6$, 1:20 ratio) were mixed in 1% adult bovine serum BI-S-33 medium containing 10 µM Igl–C or Igl–C3 protein, centrifuged at 150 *g* for 5 min at 4°C, then incubated on ice for 2 h.

In the experiment to detect the expression of WT and double-mutant Igl–C3 surface proteins in transfected CHO cells, 5% BSA in PBS was used for blocking after 4% polyformaldehyde fixation. EH3077 (1:100) was used as the primary antibody, and Alexa Fluor 594 goat anti-mouse IgG H&L (1:200; Invitrogen, A11005) was used as the secondary antibody. Flow cytometry was performed with a BD FACSCalibur Flow Cytometer.

## Laser confocal microscopy

To determine the expression of WT, double-mutant and vector control EGFP-Igl-C3 proteins on adherent and suspended CHO cell surface, CHO cells were divided into two groups, one of which was seeded on a glass-bottom dish (Cellvis, D35C4-20-1.5-N; Mountain View, CA, USA), and the other was in suspension. Before blocking with 5% BSA in PBS, the cells were fixed with 4% paraformaldehyde carefully without any detergent for permeabilization. EH3077 (1:50) was used as the primary antibody, while Alexa Fluor 594 goat anti-mouse IgG H&L (1:200; Invitrogen, A11005) was used as the secondary antibody. Suspended CHO cells were then smeared on cytoslides with a Cytospin 4 Cytocentrifuge (Thermo Scientific, Waltham, MA, USA). The dishes and cytoslides were observed under a laser confocal microscope (SP8, Leica Microsystems, Wetzlar, Germany).

## Western blotting

Western blotting was implemented to examine the transfection of WT and double-mutant Igl–C3 surface proteins into CHO cells. Protein samples from cells transfected with various plasmids were pelleted by centrifugation, separated on 12.5% polyacrylamide gels, then electrotransferred onto polyvinylidene difluoride membranes (General Electric Co., Schenectady, NY, USA). After blocking with 5% BSA in PBS, membranes were incubated with the following primary antibodies: GFP (D5.1) Rabbit mAb (1:2,000; Cell Signaling Technology, #2956; Boston, MA, USA) and anti-β-actin antibody (1:2,000; Abcam, Ab8227; Cambridge, United Kingdom). The secondary antibody was HRP-conjugated goat anti-rabbit IgG H&L (1:4 000; Abcam, Ab6721; Cambridge, United Kingdom).

The proteins were finally detected using an enhanced chemiluminescence (ECL) Western Blotting Substrate Kit (Tanon, Shanghai, China).

## Animal model for amebic liver abscess

Six-week-old male hamsters were obtained from Shanghai Songlian Experimental Animal Factory. ALA was induced as previously described (18). In brief, $1 \times 10^6$ axenic *E. histolytica* SAW755CR trophozoites were inoculated directly into the liver. After euthanizing the hamsters at day 7 post-infection, liver tissues were harvested, fixed with 4% paraformaldehyde, and embedded in paraffin. To detect the distribution of Igl protein in the liver, immunohistochemical staining was performed using EH3077 as the primary antibody and HRP-conjugated goat anti-mouse IgG H&L (Abcam, Ab6789; Cambridge, United Kingdom) as the secondary antibody.

## Cell viability assay

Quantitative determination of ATP was performed to detect the number of living Caco-2 cells in 96-well plates using a CellTiter-Glo luminescent cell viability assay kit (Promega, Madison, WI, USA). After incubation with eukaryote-expressed recombinant *E. histolytica* Igls (200 nM) for 0, 4, 8, and 12 h, the CellTiter-Glo reagent was added into wells, which were then incubated at room temperature for 10 min. Luminescent intensity was measured using a FlexStation 3 Multi-Mode Microplate Reader (Molecular Devices, MD, USA).

## Quantitative real-time RT-PCR

RAW264.7 cells were preincubated with eukaryote-expressed recombinant *E. histolytica* proteins (200 nM) before harvest. Total RNA from these cells was purified with an RNeasy Plus Mini Kit (Qiagen, Dusseldorf, Germany), then used for reverse transcription by a PrimeScript 1st strand cDNA Synthesis Kit (TaKaRa, Shiga, Japan). In accordance with manufacturer's protocols, qRT-PCR was conducted in a final reaction volume of 20 µL on an ABI 7500 real-time PCR system (Applied Biosystems, CA, USA), containing primers for IL-1β, IL-6, TNF-α, iNOS, and GAPDH genes (see Table S4). With a SYBR Premix Ex Taq (TaKaRa, Shiga, Japan), reactions were performed in 96-well plates under the following amplification cycling conditions: 30 s at 95℃, 40 cycles of 5 s at 95℃, and 35 s at 60℃. Gene expressions were analyzed using the $2^{-\Delta\Delta Ct}$ method.

## Statistical analysis

GraphPad Prism software version 8.3.0 was used for statistical analysis. All data were analyzed using Student's *t*-test or one-way analysis of variance (two-tailed), while Bonferroni correction was applied on the significance of multiple testing. $P < 0.05$ was considered statistically significant.

## ACKNOWLEDGMENTS

We thank all members of the laboratory for fruitful discussions and constructive suggestions.

This study was supported by the National Key Research and Development Program of China (2018YFA0507304) and National Natural Science Foundation of China (81630057).

## AUTHOR AFFILIATIONS

[1]Department of Medical Microbiology and Parasitology, School of Basic Medical Sciences, Fudan University, Shanghai, China
[2]Department of Infectious Diseases, Tokai University School of Medicine, Isehara, Kanagawa, Japan

## AUTHOR ORCIDs

Hiroshi Tachibana ⓘ http://orcid.org/0000-0002-1807-9090
Xunjia Cheng ⓘ http://orcid.org/0000-0002-8851-6903

## FUNDING

| Funder | Grant(s) | Author(s) |
|---|---|---|
| MOST \| National Key Research and Development Program of China (NKPs) | 2018YFA0507304 | Xunjia Cheng |
| MOST \| National Natural Science Foundation of China (NSFC) | 81630057 | Xunjia Cheng |

## AUTHOR CONTRIBUTIONS

Hongze Zhang, Data curation, Formal analysis, Validation, Writing – original draft | Qingshan Li, Data curation, Formal analysis.

## ETHICS APPROVAL

Fresh human blood group O was obtained from healthy voluntary donors, being used in hemagglutinating, hemolytic, erythrocytic phagocytic, and adherence assays. The procedure for obtaining fresh blood from donors was carried out in accordance with the international guidelines for the study in human populations of the Declaration of Helsinki.

In strict accordance with the guidelines of the Regulations for the Administration of Affairs Concerning Experimental Animals (1988.11.1), all animal experiments were conducted with the approval of the Institutional Animal Care and Use Committee (permit no. 20160225–097). All efforts were made to minimize animal suffering.

## ADDITIONAL FILES

The following material is available online.

### Supplemental Material

**Supplemental material (Spectrum00538-24-s0001.pdf).** Fig S1 to S5; Table S1 to S4.

### Open Peer Review

**PEER REVIEW HISTORY (review-history.pdf).** An accounting of the reviewer comments and feedback.

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
