## [Reviewer comments · Microbiology Spectrum]

Microbiology Spectrum

Identification and characterization of a carbohydrate recognition domain-like region in *Entamoeba histolytica* Gal/GalNAc lectin intermediate subunit

Hongze Zhang, Qingshan Li, Hang Zhou, Meng Feng, Yanqing Zhao, Ruixue Zhou, Lijun Chen, Hiroshi Tachibana, and Xunjia Cheng

Corresponding Author(s): Xunjia Cheng, Fudan University School of Basic Medical Sciences

Review Timeline:

Submission Date:	February 29, 2024
Editorial Decision:	June 3, 2024
Revision Received:	July 2, 2024
Editorial Decision:	July 30, 2024
Revision Received:	August 9, 2024
Accepted:	September 6, 2024

Editor: Galadriel Hovel-Miner

Reviewer(s): Disclosure of reviewer identity is with reference to reviewer comments included in decision letter(s). The following individuals involved in review of your submission have agreed to reveal their identity: Lesly A Temesvari (Reviewer #1)

Transaction Report:

DOI: <https://doi.org/10.1128/spectrum.00538-24>

Re: Spectrum00538-24 (Identification and characterization of a carbohydrate recognition domain-like region in *Entamoeba histolytica* Gal/GalNAc lectin intermediate subunit)

Dear Prof. Xunjia Cheng:

Thank you for the privilege of reviewing your work. Below you will find my comments, instructions from the Spectrum editorial office, and the reviewer comments.

Revision Guidelines

Sincerely,
Galadriel Hovel-Miner
Editor
Microbiology Spectrum

Reviewer #1 (Comments for the Author):

This is a study by Zhang et al., that characterizes the lectin activity of Igl, a cell surface protein in *Entamoeba histolytica*, the causative agent of amoebic dysentery and liver abscess. The study has several notable strengths. First, the work will be of high importance to the *Entamoeba* community of researchers as it is the first to extensively characterize the carbohydrate recognition domain (CRD) of Igl. Second, given the comprehensiveness of the study, the work will be of interest to researchers outside the

Entamoeba community, because others will be able to model studies of lectins from other systems. Third, given that Igl is thought to participate in virulence, and given that Igl is unique to the parasite (not in the host) the work provides insight into pathogenicity and into a protein that may serve as a vaccine target. The authors have responded well to the other critiques- especially the comments about cell surface localization of Igl in the CHO-heterologous system. This reviewer has the following minor comments:

1. It is not clear to this reviewer how and why the data differs between Figs. 3C and 4C. Is it measuring the same function (i.e., uptake of Jurkat)? If so, the data values (Y-axis) are quite different.
2. The images of the well plates do not seem to reflect the data presented in the hemagglutination assays. However, this reviewer appreciates that the authors are including these images and is confident in the numerical data. This was a comment in previous reviews, but this reviewer realizes that this might not be able to be rectified because it may just be a result of image taken from above and angular distortion.

Reviewer #2 (Comments for the Author):

The revised paper addresses the critiques of the previous reviewers. That said, the present reviewer has serious concerns about the methods used here, which are those of the late 20th century rather than the 21st century. In particular, the authors used methods, which were previously used to identify the binding domain of the large subunit of the Entamoeba Gal/GalNAc lectin (that is called LecA), to identify the Gal/GalNAc-binding domain of the intermediate subunit.

- 1) They take advantage of a monoclonal antibody made by others, which inhibits binding of the intermediate subunit to host cells, to map Cys-rich segments of the intermediate lectin (a good idea).
- 2) The first problem, which is crippling, is that the segments of the intermediate subunit are made in the cytosol of bacteria, where disulfides are not formed. Instead, disulfides are made in the periplasm of bacteria, which is equivalent to the secretory system of eukaryotic cells.
- 3) The second problem is that Cys to Ser mutations in the intermediate subunit just create havoc by cross-linking the cytosol-expressed proteins.
- 4) The third problem is that they use sequence-based alignments to compare the intermediate subunit with Gal-binding lectins of the host, which are not informative.
- 5) What they need to do to solve the problems is to use 21st century methods including AlphaFold to predict the structure of the intermediate subunit, which is available at NCBI or AmoebaDB. The structure by itself will identify disulfide bonds, as well as identify charged and/or amino acids on its surface. Comparison of the structure to that of host Gal-binding lectins, which have been extensively studied, will then suggest amino acids important for binding Gal/GalNAc that might be tested by mutations to Ala.

In this manuscript the authors investigate the presences of an adhesion related region on the intermediate subunit (Igl) of the Gal/GalNAc-inhibitable lectin. This is significant because Igl has previously been shown to be involved in adhesion, but to date, Igl has not been shown to have a CRD. By using recombinant Igl proteins representing different segments of full-length Igl and recombinant segments containing point mutations, the authors identify an adhesion related region of the Igl subunit. Experiments also examine its role in hemagglutinating activity, RBC attachment and phagocytosis of Jurkat cells. Finally, they examine its hemolytic activity and its effect on cell viability and production of pro-inflammatory cytokines. Overall, this paper appears to be well thought out and many functional aspects of the Igl subunit are tested. The discussion section is also well written and contains a lot of relevant information. However, conclusions rely strongly on traditional flowcytometry to quantify attachment and phagocytosis which is problematic (see below).

- 1) There are minor grammatical errors present throughout the manuscript that require attention. It would benefit from further editing.
- 2) Fig S3: This figure characterizes the ability of EH3077 to inhibit adhesion and phagocytosis of RBCs. While the representative images in A and D do appear to show this, by using microscopy alone it is impossible to differentiate between phagocytosed cells and cells that are simply adhered to the outside of the amoebae but localized in the center instead of on the edges.
- 3) Fig 2: It can be assumed that the experiments performed in this figure use recombinant Igl proteins that do not contain the point mutations and that recombinant proteins that do include the point mutations are introduced in later figures. However, this is not entirely clear from the text.
- 4) Traditional flow cytometry alone is not entirely adequate for quantifying adherence and phagocytosis of Jurkat cells. Detecting the fluoresces of both FITC labeled amoebae and DiD labeled Jurkat cells together does not indicate localization and cannot differentiate between attached Jurkat cells and ingested cells. While the assays are set up in such a way that it would be expected that incubation of trophozoites with heat-killed Jurkat cells at 37 degrees C would result in phagocytosis, and that incubation on ice would inhibit ingestion and result in attachment, the authors do not convincingly show that this is true of their results.

The same comments apply to the use of traditional flow cytometry to measure adherence of trophozoites to CHO cells.

- 5) Fig: 3+4, F+G: The differences in the APC+ FITC+ quadrant between the control and Igl-C incubated conditions shown in F and G do not appear to be significant. It is also not clear

how the percentages shown in G are calculated when looking at the representative images shown in F.

- 6) Fig: 6 A+B: It is not at first apparent that EH3077 is being used as the primary antibody in this figure. Perhaps making this information more readily available in the results section or the figure legend would help with interpretation.

Response to Reviewer 1:

This is a study by Zhang et al., that characterizes the lectin activity of Igl, a cell surface protein in *Entamoeba histolytica*, the causative agent of amoebic dysentery and liver abscess. The study has several notable strengths. First, the work will be of high importance to the *Entamoeba* community of researchers as it is the first to extensively characterize the carbohydrate recognition domain (CRD) of Igl. Second, given the comprehensiveness of the study, the work will be of interest to researchers outside the *Entamoeba* community, because others will be able to model studies of lectins from other systems. Third, given that Igl is thought to participate in virulence, and given that Igl is unique to the parasite (not in the host) the work provides insight into pathogenicity and into a protein that may serve as a vaccine target. The authors have responded well to the other critiques-especially the comments about cell surface localization of Igl in the CHO-heterologous system. This reviewer has the following minor comments:

1. It is not clear to this reviewer how and why the data differs between Figs. 3C and 4C. Is it measuring the same function (i.e., uptake of Jurkat)? If so, the data values (Y-axis) are quite different.

Reply: Thank you very much for your thoughtful comment. In fact, this is exactly an important issue in the process of *E. histolytica* in vitro culture. The pathogenicity of *E. histolytica* trophozoites is unstable during in vitro culture (the pathogenicity usually decreases and requires re-poisoning through passing hamster livers), and their biological characteristics are greatly affected by the proliferation status (such as trophozoite phagocytosis and adherence). In the present study, the repeated phagocytic inhibition experiments in Figure 2D and Figure 4C were conducted in close proximity, while those experiments in Figure 3C were conducted after a few weeks. Though we tried to keep their experimental operations strictly consistent, the overall levels of trophozoite phagocytosis did vary.

2. The images of the well plates do not seem to reflect the data presented in the hemagglutination assays. However, this reviewer appreciates that the authors are including these images and is confident in the numerical data. This was a comment in previous reviews, but this reviewer realizes that this might not be able to be rectified because it may just be a result of image taken from above and angular distortion.

Reply: Thank you for the valuable comment. Your understanding and support are very important to us.

Response to Reviewer 2:

The revised paper addresses the critiques of the previous reviewers. That said, the present reviewer has serious concerns about the methods used here, which are those of the late 20th century rather than the 21st century. In particular, the authors used methods, which were previously used to identify the binding domain of the large subunit of the *Entamoeba* Gal/GalNAc lectin (that is called LecA), to identify the Gal/GalNAc-binding domain of the intermediate subunit.

1. They take advantage of a monoclonal antibody made by others, which inhibits binding of the intermediate subunit to host cells, to map Cys-rich segments of the intermediate lectin (a good idea).

Reply: Thank you for your encouragement. At first, we prepared a series of monoclonal antibodies over 20 years ago (including EH3077 used in the present study), then used them to identify the *E. histolytica* Gal/GalNAc lectin intermediate subunit (1). Years later, we found this inhibitory phenomenon in follow-up experiments. In fact, this was precisely an important clue to our initial speculation that *E. histolytica* Igl had a CRD-like region.

2. The first problem, which is crippling, is that the segments of the intermediate subunit are made in the cytosol of bacteria, where disulfides are not formed. Instead, disulfides are made in the periplasm of bacteria, which is equivalent to the secretory system of eukaryotic cells.

Reply: Thank you for highlighting this issue. This is indeed a very important question that got us thinking a lot when designing experiments. We understand that within bacteria, the formation of disulfide bonds in proteins only occurs at the specific site, that is the highly oxidizing environment of the bacterial periplasm. In previous work, we have used the pET system to express some recombinant Igl fragments many times with high protein expression level and good patient serum reactivity (2, 3). In experimental operations, however, though the pET-19b vector does not have a signal sequence, we found that a small number of its transformed expression products (using *E. coli* BL21 Star(DE3)pLysS) still exist in soluble form. After the detection with

monoclonal antibodies and patient serum, the biological properties of these soluble Igl products were found to be consistent with those of natural Igl proteins. Therefore, we believe that while most Igl fragment proteins exist as inclusion bodies in the cytosol, there are quite a few proteins that can behave as soluble in the periplasm and form correct disulfide bonds. Moreover, we used to use pET-19b vector to express *E. histolytica* peroxiredoxin protein, which products were even basically soluble under the same experimental conditions (4). In other publications using the pET-28a vector (no signal sequence either), some protein products are also soluble or partially soluble, which is consistent with our results (5, 6).

Even if the protein folding can be performed correctly, the prokaryotic system does have the defect of not being able to undergo post-translational modifications (like glycosylation). This limitation should be clearly pointed out in the manuscript, and corresponding contents have been emphasized in the Discussion section (page 20, line 382-387).

3. The second problem is that Cys to Ser mutations in the intermediate subunit just create havoc by cross-linking the cytosol-expressed proteins.

Reply: Thank you for the comment. As described in the manuscript, the first cysteine in each selected CXXC motif was actually converted into glycine, not serine (see page 8, line 137; page 41, line 830). We again consulted the relevant literatures, finding that the mutation to serine can lead to cross-linking, while the mutation to glycine has no such serious effect. Considering the amino acid properties (like acid-base), we mutated cysteine to glycine, but we also agree that the mutation to alanine you mentioned later is a good alternative.

4. The third problem is that they use sequence-based alignments to compare the intermediate subunit with Gal-binding lectins of the host, which are not informative.

Reply: Thank you for the thoughtful comment. Indeed, the overall similarity between Igl CRD-like region and mammalian galectin-2 proteins was low. This is also a big difficulty in our research process: the similarity of *E. histolytica* Igl or Hgl to lectins of other species is very low, which makes it difficult to predict or detect the structure of Igl protein for identifying its potential carbohydrate-binding sites (lack of appropriate references). Here, we simply found that the two cysteine residues remain

highly conserved even when the overall similarity is not high. The corresponding statement has been adjusted in the manuscript (page 11, line 206-208).

5. What they need to do to solve the problems is to use 21st century methods including AlphaFold to predict the structure of the intermediate subunit, which is available at NCBI or AmoebaDB. The structure by itself will identify disulfide bonds, as well as identify charged and/or amino acids on its surface. Comparison of the structure to that of host Gal-binding lectins, which have been extensively studied, will then suggest amino acids important for binding Gal/GalNAc that might be tested by mutations to Ala.

Reply: Thank you for your critical suggestion. Actually, we have tried two methods to detect the structure of native *E. histolytica* Igl protein, but without success. ① Cryo-electron microscopy: Under 200-kV cryo-TEMs, the particles of Igl were not uniform enough and the overall profile was not clear, which could not meet the requirements of further research using 300-kV cryo-TEMs. To confirm whether the molecular weight is too low, we also used Igl protein bound with antibodies for experiments, but the effect was not significantly improved, and there existed many irregular polymers after the addition of antibodies. ② X-ray diffraction: The method requires protein crystals, but we failed to successfully crystallize Igl alone. Our current analysis suggests that these difficulties may be due to the properties of such membrane proteins and the complex spatial structure caused by disulfide bonds of excessive cysteines, which also results in Igl protein's low model confidence in its AlphaFold structure prediction. The low similarity mentioned in the previous reply makes the structural prediction and analysis more difficult as well. In this study, we basically used the traditional methods of *E. histolytica* research, and made a small improvement by referring to the saccharide binding assay in the Gal/GalNAc lectin identification of *Fusobacterium nucleatum* (7). Our experimental method is indeed a major limitation, which should be emphasized in the Discussion section (added on page 20, line 387-396).

Response to Reviewer 3:

In this manuscript the authors investigate the presences of an adhesion related region on the intermediate subunit (Igl) of the Gal/GalNAc-inhibitable lectin. This is significant because Igl has previously been shown to be involved in adhesion, but to

date, Igl has not been shown to have a CRD. By using recombinant Igl proteins representing different segments of full-length Igl and recombinant segments containing point mutations, the authors identify an adhesion related region of the Igl subunit. Experiments also examine its role in hemagglutinating activity, RBC attachment and phagocytosis of Jurkat cells. Finally, they examine its hemolytic activity and its effect on cell viability and production of pro-inflammatory cytokines. Overall, this paper appears to be well thought out and many functional aspects of the Igl subunit are tested. The discussion section is also well written and contains a lot of relevant information. However, conclusions rely strongly on traditional flowcytometry to quantify attachment and phagocytosis which is problematic (see below).

1. There are minor grammatical errors present throughout the manuscript that require attention. It would benefit from further editing.

Reply: Thank you very much for your valuable comment. The manuscript has now been corrected by a professional English proofreader (Editage: www.editage.cn), with the corresponding changes recorded in the “Marked Up Manuscript” file.

2. Fig S3: This figure characterizes the ability of EH3077 to inhibit adhesion and phagocytosis of RBCs. While the representative images in A and D do appear to show this, by using microscopy alone it is impossible to differentiate between phagocytosed cells and cells that are simply adhered to the outside of the amoebae but localized in the center instead of on the edges.

Reply: Thank you for your critical comment. In the phagocytosis experiment in this figure, distilled water was quickly added to dissolve the free erythrocytes after co-incubation at 37 degrees, then fixed and stained. Therefore, only ingested erythrocytes are present, while free and edge-adhering ones are not (see Figure S3D). However, in adherence experiments, the coincidence in the position you mentioned is possible. At present, we try to solve this problem mainly by changing the focusing plane of the microscope.

To avoid such interference, we further tried to use flow cytometry for fluorescence detection: the Flow Cytometer instrument tends to prepare each sample as single cells (based on its working mode), allowing host cells that are adhered but not engulfed to

be separated from the trophozoites (see Figure 2C, 3B, and 4B). Thus, in these amoebic adherence experiments, flow cytometry will also break the adhesion and interfere with the results (see Figure 3F and 4F, described on page 11, line 194-196). We do still need to find an objective and effective experimental method to detect *E. histolytica* adherence.

3. Fig 2: It can be assumed that the experiments performed in this figure use recombinant Igl proteins that do not contain the point mutations and that recombinant proteins that do include the point mutations are introduced in later figures. However, this is not entirely clear from the text.

Reply: Thank you for the thoughtful suggestion. Yes, the Igl fragments used in Figure 2 are without mutations, while the mutated Igl fragments are used in Figure 3 and Figure 4. It was not written clearly in the manuscript before, and now an explanatory statement has been added (page 9, line 150-151).

4. Traditional flow cytometry alone is not entirely adequate for quantifying adherence and phagocytosis of Jurkat cells. Detecting the fluoresces of both FITC labeled amoebae and DiD labeled Jurkat cells together does not indicate localization and cannot differentiate between attached Jurkat cells and ingested cells. While the assays are set up in such a way that it would be expected that incubation of trophozoites with heat-killed Jurkat cells at 37 degrees C would result in phagocytosis, and that incubation on ice would inhibit ingestion and result in attachment, the authors do not convincingly show that this is true of their results. The same comments apply to the use of traditional flow cytometry to measure adherence of trophozoites to CHO cells.

Reply: Thank you for your critical comment. Traditionally, microscopic counting is performed after co-incubation of *E. histolytica* trophozoites and cells at 37 degrees (phagocytosis) or 0 degrees (adherence) (1). As mentioned in the above reply, we have found in practice that the application of flow cytometry is more successful in phagocytosis experiments, but it will lead to larger deviations in adherence experiments. We have tried to observe the parasite adherence with laser confocal microscopy, which also cannot quantify the results well. Thus, new objective and effective methods are still needed.

5. Fig: 3+4, F+G: The differences in the APC+ FITC+ quadrant between the control

and Igl-C incubated conditions shown in F and G do not appear to be significant. It is also not clear how the percentages shown in G are calculated when looking at the representative images shown in F.

Reply: Thank you for the comment. Yes, there is no significant difference between the groups in Figure 3G and 4G, which we think is mainly due to the working mode of flow cytometry as mentioned earlier (preparing each sample as single cells). The corresponding statement is in the manuscript (page 11, line 194-198). In Figure 3F and 4F, the positive rate of amoeba adhesion was determined by calculating the ratio of Q2/(Q2+Q3) quadrants. Q1+Q4 quadrants: CHO cells only. Q3 quadrant: amoebic trophozoites only. Q2 quadrant: amoebic trophozoites with CHO cells attached. It was not written clearly in the manuscript before, and now the description has been added (page 11, line 192-194).

6. Fig: 6 A+B: It is not at first apparent that EH3077 is being used as the primary antibody in this figure. Perhaps making this information more readily available in the results section or the figure legend would help with interpretation.

Reply: Thank you for your thoughtful suggestion. The description in the manuscript was indeed not clear enough, and the corresponding information has been emphasized in the Figure Legends section (page 43, line 888-891).

References

1. Cheng XJ, Tsukamoto H, Kaneda Y, Tachibana H. 1998. Identification of the 150-kDa surface antigen of *Entamoeba histolytica* as a galactose- and N-acetyl-D-galactosamine-inhibitable lectin. *Parasitol Res* 84:632-9.
2. Tachibana H, Cheng XJ, Masuda G, Horiki N, Takeuchi T. 2004. Evaluation of recombinant fragments of *Entamoeba histolytica* Gal/GalNAc lectin intermediate subunit for serodiagnosis of amebiasis. *J Clin Microbiol* 42:1069-74.
3. Min XY, Feng M, Guan Y, Man SQ, Fu YF, Cheng XJ, Tachibana H. 2016. Evaluation of the C-Terminal Fragment of *Entamoeba histolytica* Gal/GalNAc Lectin Intermediate Subunit as a Vaccine Candidate against Amebic Liver Abscess. *Plos Neglected Tropical Diseases* 10.
4. Li X, Zhang Y, Zhao Y, Qiao K, Feng M, Zhou H, Tachibana H, Cheng X. 2020. Autophagy Activated by Peroxiredoxin of *Entamoeba histolytica*. *Cells* 9.
5. Long X, Gou Y, Luo M, Zhang S, Zhang H, Bai L, Wu S, He Q, Chen K, Huang A, Zhou J, Wang D. 2015. Soluble expression, purification, and characterization of active recombinant human tissue plasminogen activator by auto-induction in *E. coli*. *BMC Biotechnol* 15:13.
6. Ahmed A, Fujimura NA, Tahir S, Akram M, Abbas Z, Riaz M, Raza A, Abbas R, Ahmed N. 2024. Soluble and insoluble expression of recombinant human interleukin-2 protein using pET expression vector in *Escherichia coli*. *Prep Biochem Biotechnol* doi:10.1080/10826068.2024.2361146:1-13.
7. Abed J, Emgard JE, Zamir G, Faroja M, Almogy G, Grenov A, Sol A, Naor R, Pikarsky E, Atlan KA, Mellul A, Chaushu S, Manson AL, Earl AM, Ou N, Brennan CA, Garrett WS, Bachrach G. 2016. Fap2 Mediates *Fusobacterium nucleatum* Colorectal Adenocarcinoma Enrichment by Binding to Tumor-Expressed Gal-GalNAc. *Cell Host Microbe* 20:215-25.

Re: Spectrum00538-24R1 (Identification and characterization of a carbohydrate recognition domain-like region in *Entamoeba histolytica* Gal/GalNAc lectin intermediate subunit)

Dear Prof. Xunjia Cheng:

Thank you for the privilege of reviewing your work. Below you will find my comments, instructions from the Spectrum editorial office, and the reviewer comments.

Revision Guidelines

Sincerely,
Galadriel Hovel-Miner
Editor
Microbiology Spectrum

Reviewer #1 (Comments for the Author):

This is a revised paper from Zhang et al. that seeks to discover the carbohydrate recognition domain of Igl, a surface protein on the human protozoan parasite, *Entamoeba histolytica*. *E. histolytica* infections cause considerable morbidity in developing countries. Part of a heterotrimer, that also contains the well-studied Hgl and Igl subunits, the role of Igl in adhesion to host has been elusive. Given the comprehensive nature of the study and given the clear role of Igl in virulence (see Figure 7), this work

will be of high importance to the community of Entamoeba researchers. The authors have responded well to the previous critique. This reviewer has only the following very minor comments:

1. Line 100: What is meant by "medial" side of trophozoites? This is an unconventional term when discussing different sides of membrane boundaries.
2. Figs 2C, 3B, and 4B do not have X-axis labels. It is assumed that these are flow cytometry "events." How can one obtain fewer than 0 events, which seems to be the case for some of the data?
3. Figs 3F and 4F are not mentioned in the Results section.
4. Line 170: respond should be "response."
5. The authors provide images of the hemagglutination plates along with quantification data on the lengths (e.g., see 2E and F). The quantification data are important because the images are not convincing. However, in Fig 6D the authors provide a plate image without quantification (as far as this reviewer can tell). This is not consistent with the other data analyses/figures in the paper.
6. In Fig 5, the numbers above the alignments refer to amino acids in Igl-C3. But these cannot be the actual amino acid numbers in the whole protein since this region is at the C-terminal end. So when the authors say there is a conserved cysteine at C49, it is confusing because this is the segment of amino acids: 1035-1052. The authors should use the actual amino acid number in the whole protein. The same applies to the other proteins aligned here. For example, when the authors say the conserved cysteine is at C61 in Hgl, is this the actual amino acid number in the full-length protein?
7. The Fig 1 legend is inadequate. The asterisks and "+" are not defined. It is also unconventional to call the protein "prokaryotic Igl." Recombinant is a more appropriate term since Igl is not naturally expressed in prokaryotes.

Reviewer #2 (Comments for the Author):

OK as revised.

Response to Reviewer 1:

This is a revised paper from Zhang et al. that seeks to discover the carbohydrate recognition domain of Igl, a surface protein on the human protozoan parasite, *Entamoeba histolytica*. *E. histolytica* infections cause considerable morbidity in developing countries. Part of a heterotrimer, that also contains the well-studied Hgl and Igl subunits, the role of Igl in adhesion to host has been elusive. Given the comprehensive nature of the study and given the clear role of Igl in virulence (see Figure 7), this work will be of high importance to the community of *Entamoeba* researchers. The authors have responded well to the previous critique. This reviewer has only the following very minor comments:

1. Line 100: What is meant by "medial" side of trophozoites? This is an unconventional term when discussing different sides of membrane boundaries.

Reply: Thank you for your thoughtful comment. Since elongation factor 1 alpha is located in the cytoplasm of *E. histolytica* trophozoites near the cell membrane, we intended to convey that its interaction with Igl was not outside the cell membrane, but the expression of "medial side" is indeed inappropriate. Now the corresponding statement has been changed more concisely to "Under the plasma membrane of *E. histolytica* trophozoites" (page 6, line 99-101).

2. Figs 2C, 3B, and 4B do not have X-axis labels. It is assumed that these are flow cytometry "events." How can one obtain fewer than 0 events, which seems to be the case for some of the data?

Reply: Thank you very much for the valuable comment. The horizontal coordinate in Figure 2C, 3B, and 4B is APC+ fluorescence intensity. Since DiD-labeled Jurkat cells are APC+, the cell phagocytosis of *E. histolytica* trophozoites is proportional to the intensity of this fluorescence signal. However, the fluorescence intensity of a few trophozoites was less than zero, which also raised our questions at that time: Actually, the fluorescence intensity displayed in the flow cytometry data is not the true absolute fluorescence, but the measured value corrected by the instrument firmware and software. That is, the measured value is a result of the actual fluorescence value adjusted by "baseline" and "compensation". Due to the baseline setting, calculation error of the compensation channel, and measurement error of the fluorescence

intensity, for cells with little or no fluorescence, the actual fluorescence value may be less than the baseline fluorescence value, resulting in the final measurement value being less than zero after subtraction. When this happens, some flow cytometry instruments will display the fluorescence intensity as zero, while others will directly display the corresponding negative value (a BD FACSCalibur Flow Cytometer was used in the present study). Thus, for the presence of a few cells or trophozoites with fluorescence intensity less than zero, the phenomenon is common and the results are credible.

The several X-axis labels that were not clearly written have now been added (Figure 2C, 3B, and 4B).

3. Figs 3F and 4F are not mentioned in the Results section.

Reply: Thank you for the comment. We previously described them on page 10, line 184-188, which was somewhat vague and has now been corrected.

4. Line 170: respond should be "response".

Reply: Thank you for the thoughtful suggestion. The corresponding statement has been corrected in the manuscript (page 10, line 171).

5. The authors provide images of the hemagglutination plates along with quantification data on the lengths (e.g., see 2E and F). The quantification data are important because the images are not convincing. However, in Fig 6D the authors provide a plate image without quantification (as far as this reviewer can tell). This is not consistent with the other data analyses/figures in the paper.

Reply: Thank you for your critical suggestion. To be honest, since the concern that Figure 6 was too long (now 212 mm), we didn't display quantification data in Figure 6D before for saving space. The corresponding bar chart and statistical analysis have now been added (Figure 6D). Its description in the Figure Legends section has also been supplemented (page 44, line 886-891).

6. In Fig 5, the numbers above the alignments refer to amino acids in Igl-C3. But these cannot be the actual amino acid numbers in the whole protein since this region is at the C-terminal end. So, when the authors say there is a conserved cysteine at C49,

it is confusing because this is the segment of amino acids: 1035-1052. The authors should use the actual amino acid number in the whole protein. The same applies to the other proteins aligned here. For example, when the authors say the conserved cysteine is at C61 in Hgl, is this the actual amino acid number in the full-length protein?

Reply: Thank you very much for your valuable suggestion. Indeed, the previous numerical labeling was prone to confusion and misunderstanding, and now those in Figure 5A and 5B have been changed to the actual amino acid number in the full-length Igl protein. Meanwhile, corresponding descriptions in the Results (page 11, line 198-203) and Discussion (page 17, line 322-323) sections of the manuscript have been replaced, and some contents that may cause misunderstanding have been particularly explained, such as " W1046 (W65 in galectins)".

7. The Fig 1 legend is inadequate. The asterisks and "+" are not defined. It is also unconventional to call the protein "prokaryotic Igl." Recombinant is a more appropriate term since Igl is not naturally expressed in prokaryotes.

Reply: Thank you for the critical comment. Figure 1 has now been changed, with its undefined plus sign being removed and replaced with a different asterisk. The various asterisks are now defined and annotated.

The term "prokaryotic Igl" is indeed inappropriate, and the corresponding statements in multiple sections of the manuscript have already been replaced as "prokaryote-expressed recombinant Igl" (page 7, line 112-113) or "prokaryotic expression of recombinant Igl" (page 22, line 440; page 41, line 817-818).

Re: Spectrum00538-24R2 (Identification and characterization of a carbohydrate recognition domain-like region in *Entamoeba histolytica* Gal/GalNAc lectin intermediate subunit)

Dear Prof. Xunjia Cheng:

Your manuscript has been accepted, and I am forwarding it to the ASM production staff for publication. Your paper will first be checked to make sure all elements meet the technical requirements. ASM staff will contact you if anything needs to be revised before copyediting and production can begin. Otherwise, you will be notified when your proofs are ready to be viewed.

Sincerely,
Galadriel Hovel-Miner
Editor
Microbiology Spectrum

Reviewer #2 (Comments for the Author):

corrections requested by reviewer 1 to the manuscript have been made.